# Nucleolar reorganization after cellular stress is orchestrated by SMN shuttling between nuclear compartments

Shaqraa Musawi[1,2,6], Lise-Marie Donnio[1,6] ✉, Zehui Zhao[1], Charlène Magnani[1], Phoebe Rassinoux[1], Olivier Binda[1,3], Jianbo Huang [1], Arnaud Jacquier [1], Laurent Coudert[1], Patrick Lomonte [1], Cécile Martinat [4], Laurent Schaeffer [1], Denis Mottet [5], Jocelyn Côté [3], Pierre-Olivier Mari[1] & Giuseppina Giglia-Mari [1] ✉

Spinal muscular atrophy is an autosomal recessive neuromuscular disease caused by mutations in the multifunctional protein Survival of Motor Neuron, or SMN. Within the nucleus, SMN localizes to Cajal bodies, which are associated with nucleoli, nuclear organelles dedicated to the first steps of ribosome biogenesis. The highly organized structure of the nucleolus can be dynamically altered by genotoxic agents. RNAP1, Fibrillarin, and nucleolar DNA are exported to the periphery of the nucleolus after genotoxic stress and, once DNA repair is fully completed, the organization of the nucleolus is restored. We find that SMN is required for the restoration of the nucleolar structure after genotoxic stress. During DNA repair, SMN shuttles from the Cajal bodies to the nucleolus. This shuttling is important for nucleolar homeostasis and relies on the presence of Coilin and the activity of PRMT1.

The nucleolus is a nuclear membrane-less organelle with a very structured internal organization, which associates with its different functions in ribosomal biogenesis: transcription of ribosomal DNA (rDNA) and early ribosomal RNA (rRNA) maturation[1]. Nucleoli are formed around the rDNAs which are composed of tandem head-to-tail repeats and their structure is thought to be strictly dependent on the transcriptional activity of the RNA polymerase I (RNAP1)[2]. Despite a very structured organization, nucleoli are very dynamic organelles, their shape and number can vary through the cell cycle and many proteins can enter or exit the nucleolus depending on physiological processes or cellular stress responses. This organized structure can be dynamically altered by both genotoxic agents and general cellular stress[3]. For instance, drugs that alter RNAP1 transcription (i.e. cordycepin, actinomycin D, etc.) may cause nucleolar segregation at the periphery of the nucleolus into structures known as nucleolar caps. Furthermore, drugs that block rRNA processing or the topoisomerase II (i.e. doxorubicin) but do not interfere with RNAP1 transcription induce a disruption of the compact nucleolar environment and nucleolar necklaces appear[4].

Amongst different cellular stresses known to modify the nucleolar organization, UV-irradiation has the benefit of being a quick, punctual, and chemically clean method. Moreover, cells are able to repair UV-induced lesions and hence reverse their stress status. UV-induced DNA lesions are repaired by the Nucleotide Excision Repair system (NER)[5]. NER also repairs DNA helix-distorting adducts, including environmental pollutants, the oxidative-damage derived cyclopurines[6] and participates in the repair of the ROS-induced oxidized guanine (8-oxoG)[7].

[1]Pathophysiology and Genetics of Neuron and Muscle (INMG-PGNM), CNRS UMR 5261, INSERM U1315, Université Claude Bernard Lyon 1, 68008 Lyon, France. [2]Department of Medical Laboratories Technology, College of Applied Medical Sciences, Jazan University, Jazan, Saudi Arabia. [3]Faculty of Medicine, Department of Cellular and Molecular Medicine, University of Ottawa, Ottawa K1H 8M5 Ontario, Canada. [4]INSERM/UEPS UMR 861, Paris Saclay Université, I-STEM, 91100 Corbeil-Essonnes, France. [5]GIGA-Molecular Biology of Diseases, Gene Expression and Cancer Laboratory, B34 + 1, University of Liege, Avenue de l'Hôpital 1, B-4000 Liège, Belgium. [6]These authors contributed equally: Shaqraa Musawi, Lise-Marie Donnio. ✉e-mail: lise-marie.donnio@univ-lyon1.fr; ambra.mari@univ-lyon1.fr

During UV-irradiation, it has been shown that the nucleolus is not fully disrupted but nucleolar proteins (RNAP1, Fibrillarin [FBL]) and nucleolar DNA are exported to the periphery of the nucleolus (for simplicity this phase will be called "displacement") and when DNA repair is fully completed the proper nucleolar structure is restored (for simplicity this phase will be called "repositioning")[8]. Using a best-candidate approach, we recently found that structural proteins like Nuclear Myosin I (NMI) and β-Actin (ACTB) seem to play a prominent role in this process[9]. In cells depleted from NMI and ACTB, nucleolar structure is not restored and both nucleolar proteins and nucleolar DNA remain at the periphery of the nucleolus, although DNA repair is completed and transcription is resumed[9]. However, the exact mechanism of NMI and ACTB actions on nucleolar reorganization has not yet been elucidated, probably because many other molecular actors are still unknown. In order to find a complete molecular mechanism, several structural and nucleolar proteins have been scrutinized and a certain number have been found to be crucial to restore a proper nucleolar structure after DNA repair completion. One of these proteins is FBL. Consequently, we studied whether FBL interacting partners were also involved in this process. Amongst these different FBL partners, we investigated whether the protein Survival of Motor Neurons (SMN) was implicated in the restoration of the nucleolar structure after DNA repair completion.

Spinal Muscular Atrophy (SMA) is an autosomal recessive neuromuscular disease, which affects neurons that controls the voluntary movement of muscles (motoneurons)[10]. In SMA, motoneurons are progressively lost leading to progressive muscle wasting and atrophy because muscles no longer receive signals from the motor neurons of the spinal cord. Children affected with SMA have symptoms that can vary greatly depending on the age of disease onset and its severity. Normal activities, such as crawling, walking, maintaining a seated position, controlling head movements, breathing and swallowing, might be affected[10]. With an incidence of 1 in 6000–10,000 live births, SMA is the most prevalent hereditary cause of infant mortality[11].

SMA is caused by bi-allelic mutations in the *SMN1* gene (Survival of Motor Neuron: SMN) and the disease phenotype is modified by the number of copies of a second paralog gene, *SMN2*, which is always present in SMA patients[12]. *SMN1* produces a full-length functional version of the SMN protein whereas in *SMN2*, the absence of exon 7 in most of the transcripts produces an unstable version of the SMN protein (SMNΔ7). *SMN2* can express about 10–15% of the full-length protein, which is insufficient to avoid the disease.

SMN is a multifunctional protein involved in many cellular processes, such as biogenesis and trafficking of ribonucleoproteins, local translation of messenger RNAs, etc[13]. SMN protein is ubiquitously expressed and is localized to both the cytoplasm and the nucleus. Within the nucleus, SMN localizes in Gems and Cajal bodies (CBs), which have been shown to associate with nucleoli[14]. Within CBs, SMN interacts with the protein Coilin[15]. In certain conditions, SMN is also detected in the nucleoli of mammalian primary neurons and colocalized with FBL[16]. In addition, a transient colocalization of SMN at the periphery of nucleoli with FBL after actinomycin D treatment in 10–20% of Hela cells[17,18] suggests that SMN could be present in the nucleolus under stress conditions. SMN Tudor domain is involved in the binding to RGG motifs containing proteins such as FBL and Coilin[19].

We investigated the possible role of SMN in nucleolar reorganization during both displacement and repositioning of RNAP1, during and after DNA repair of UV-induced damage and generally after stress induction. We show here that in the absence of a functional SMN, both RNAP1 and FBL remain at the periphery of the nucleolus after DNA repair completion even once transcription is fully restored. We could reveal that SMN (and its complex) shuttles to the nucleolus after DNA repair completion. We determine that this shuttling is dependent on the presence of Coilin and governed by PRMT1-dependent arginine methylation activity. Additionally, we disclosed that the presence of

FBL is important for the proper restoration of SMN into CBs. We could show that SMN cells show a sensitivity to DNA damage and in particular, to chronic oxidative damage. Our results demonstrate a role for SMN in nucleolar homeostasis with potential implications for SMA pathology.

## Results

### RNAP1 and FBL repositioning after DNA repair completion are SMN-dependent

To investigate a possible role of SMN in nucleolar reorganization in response to cellular stress, we investigated whether the previously reported[8] RNAP1 UV-induced displacement and the later repositioning were still happening in the absence of SMN. As SMN-deficient cells we used primary fibroblasts from SMA patients (Fig. S1a), SMA iPSC-derived motoneurons (Fig. S2) and transformed fibroblasts in which SMN was downregulated by lentiviral transfection of 2 independent inducible shRNAs against SMN 3'UTR (Fig. 1c). Using these cell lines, we performed immunofluorescent (IF) assays to detect both RNAP1 and FBL positioning in the absence of damage (No UV), 3 h post-UV-irradiation (PUVI) (this time point corresponds to the minimum of RNAP1 transcriptional activity as found in[8] and at 40 h PUVI (this time point corresponds to the RNAP1 full recovery of transcriptional activity and full DNA repair as described in[8]). Wild-type fibroblasts, SV40-transformed MRC5 and primary C5RO were used as positive controls, while Cockayne Syndrome type B (CSB) TC-NER deficient fibroblasts (both transformed and primary; termed CS1AN) were used as negative control, as used in[8]. For iPSC-derived motoneurons, WT and SMA iPSC-derived neuronal progenitors were differentiated into motoneurons[20].

As described in[8], UV-irradiation induced a displacement of both RNAP1 and FBL to the periphery of nucleoli in all cell lines tested (Fig. 1a, b, Figs. S1b, c, S2 at 3 h PUVI). As expected, in wild-type cells (MRC5, shSCRAMBLE and C5RO) both RNAP1 and FBL recovered their position within the nucleoli at 40 h PUVI (Fig. 1a, b, Fig. S1b, c). In contrast, in cells depleted of SMN (Fig. 1a, b, Sh5-SMN and Sh6-SMN), primary fibroblasts mutated in *SMN1* (Fig. S1b, c, SMA1) and SMA iPSC-derived motoneurons (Fig. S2a, b) neither RNAP1 nor FBL recovered the proper position within the nucleoli after DNA repair completion. As previously demonstrated[8], in CSB-deficient cells no return of the RNAP1 and FBL was observed (Fig. 1a, b, Fig. S1b, c, CS1AN).

In CSB-deficient cells, the repositioning of RNAP1 and FBL is impeded because DNA lesions on the transcribed strand of rDNA genes are not properly repaired and RNAP1 transcription is not restored[8]. To investigate whether this was the case in SMN-deficient cells, we performed an RNA-fish assay detecting the pre-rRNA transcript using a specific probe against the 47 S product (Fig. S3a) and could determine (Fig. S3b, S1e) and quantify (Fig. 1d, S1d) that RNAP1 transcription is restored in SMN-deficient cells at 40 h PUVI as in wild-type cells. In parallel, the involvement of SMN in Nucleotide Excision Repair (NER) was studied by performing UDS (Fig. S4a), RRS (Fig. S4b) and TCR-UDS (Fig. S4c, d) experiments in cells depleted of SMN. Our results clearly show that SMN has no role in NER (Fig. S4).

Taken together, these results indicate that in the absence of SMN, RNAP1 and FBL are correctly displaced at the periphery of the nucleolus in response to DNA damage but are not repositioned within the nucleolus once DNA repair reactions are completed and RNAP1 transcription is restored.

### SMN-complex shuttles in the nucleolus and co-localizes with nucleolar proteins after UV irradiation

We showed that SMN is required for the proper repositioning of RNAP1 and FBL at late time points PUVI. We questioned how SMN could be involved in this mechanism if, usually, it is not present in the nucleoli. In fact, SMN protein is typically located in the cytoplasm and within the nucleus where SMN is found in CBs together with Coilin and in Gems

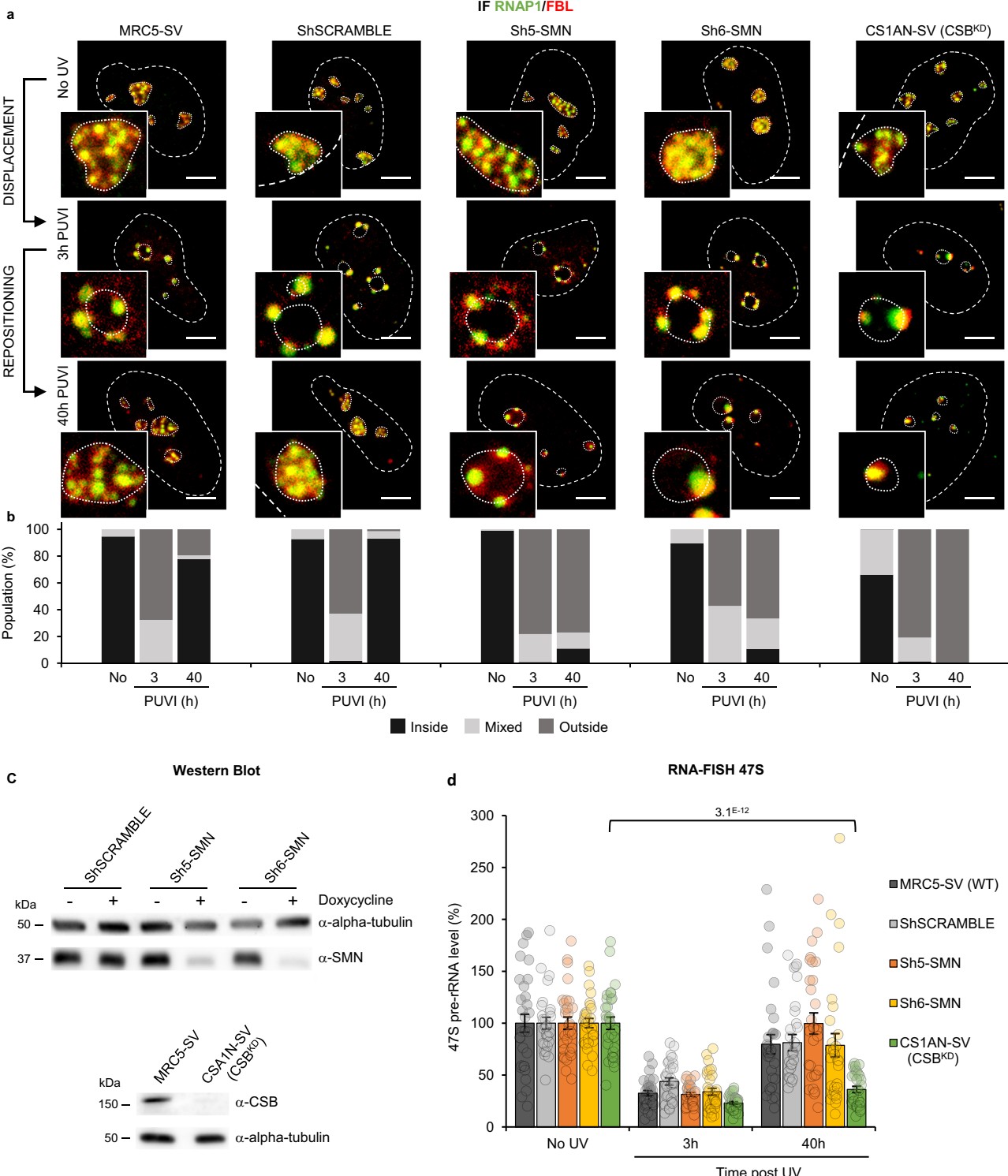

**Fig. 1 | RNAP1 and FBL localization during DNA repair in SMN deficient transformed fibroblasts. a** Representative confocal microscopy images of immunofluorescence (IF) assay in MRC5 cells showing, after 16 J/m² UV-C irradiation, the localization of RNAP1 (green) and FBL (red) in transformed fibroblasts, at different times Post UV-Irradiation (PUVI). Nuclei and nucleoli are indicated by dashed and dotted lines respectively. Scale bar: 5 μm. **b** Quantification of cell number for RNAP1 localization (inside the nucleolus, outside the nucleolus or mixed localization) at different times PUVI. At least 50 cells from one representative experiment were analyzed. **c** Western Blot of SMN and CSB on whole cell extracts of transformed fibroblasts (MRC5-SV cells). Doxycycline treatment induces the expression of the ShRNA against SMN. **d** Quantification of RNA-FISH assay showing the 47 S pre-rRNA level after UV-C exposure in transformed fibroblasts. Data are represented as mean values +/− SEM. At least 27 cells was quantified from one representative experiment. The *p*-value correspond to a student's test with two-tailed distribution and two-sample unequal variance to compare after irradiation with No UV condition. Source data of uncropped gel and graphs are provided as a Source Data file.

without Coilin. To study the localization of SMN and RNAP1 during the DNA repair process, we initially performed immunofluorescence assays at 3 h and 40 h PUVI in wild-type cells. At 40 h PUVI, two co-existing populations of cells could be detected: (i) a majority of cells in which RNAP1 is repositioned within the nucleolus and SMN is localized in the CBs/Gems and (ii) a minority of cells in which RNAP1 is still localized at the periphery of the nucleolus and SMN is unusually localized at the periphery of the nucleolus and cannot be detected in CBs/Gems anymore (Fig. S5a–c). Because of this result, we decided to extend our analysis by adding a time point intermediate between 3 h and 40 h PUVI (24 h PUVI) and a time point beyond the 40 h PUVI (48 h PUVI) and performed the IF assays against SMN and different nucleolar proteins or SMN protein partners (Fig. 2). We verified the RNAP1 transcriptional activity by RNA-FISH of 47 S at 24 h and 48 h PUVI and show that RNAP1 transcription is not yet restored at 24 h PUVI but it's

fully recovered at 48 h PUVI (Fig. S5d). At 24 h PUVI, our results revealed the presence of SMN at the periphery of and/or within the nucleolus in the vast majority of cells (Fig. 2a, quantification in 2d). Concomitantly, at 24 h PUVI, RNAP1 was found to be localized at the periphery of the nucleolus (Fig. 2a, quantification in 2e). On the other hand, we observed a complete return to the undamaged condition (RNAP1 within the nucleolus and SMN in the CBs) at 48 h PUVI in the vast majority of cells (Fig. 2a, quantification in 2d and 2e). Despite the presence of SMN and RNAP1 at the nucleolar periphery together at 24 h post-UV, no co-localization between these two proteins was observed (Fig. 2a, panel 24 h PUVI).

We showed that the loss of SMN alters FBL localization at 40 h PUVI (Figs. 1a, S1b). Using GST pull-down assays we confirmed that FBL from cell extracts interacts with SMN[17,21,22] (Fig. S5e). Furthermore, using a panel of SMA-linked TUDOR domain mutants, we established

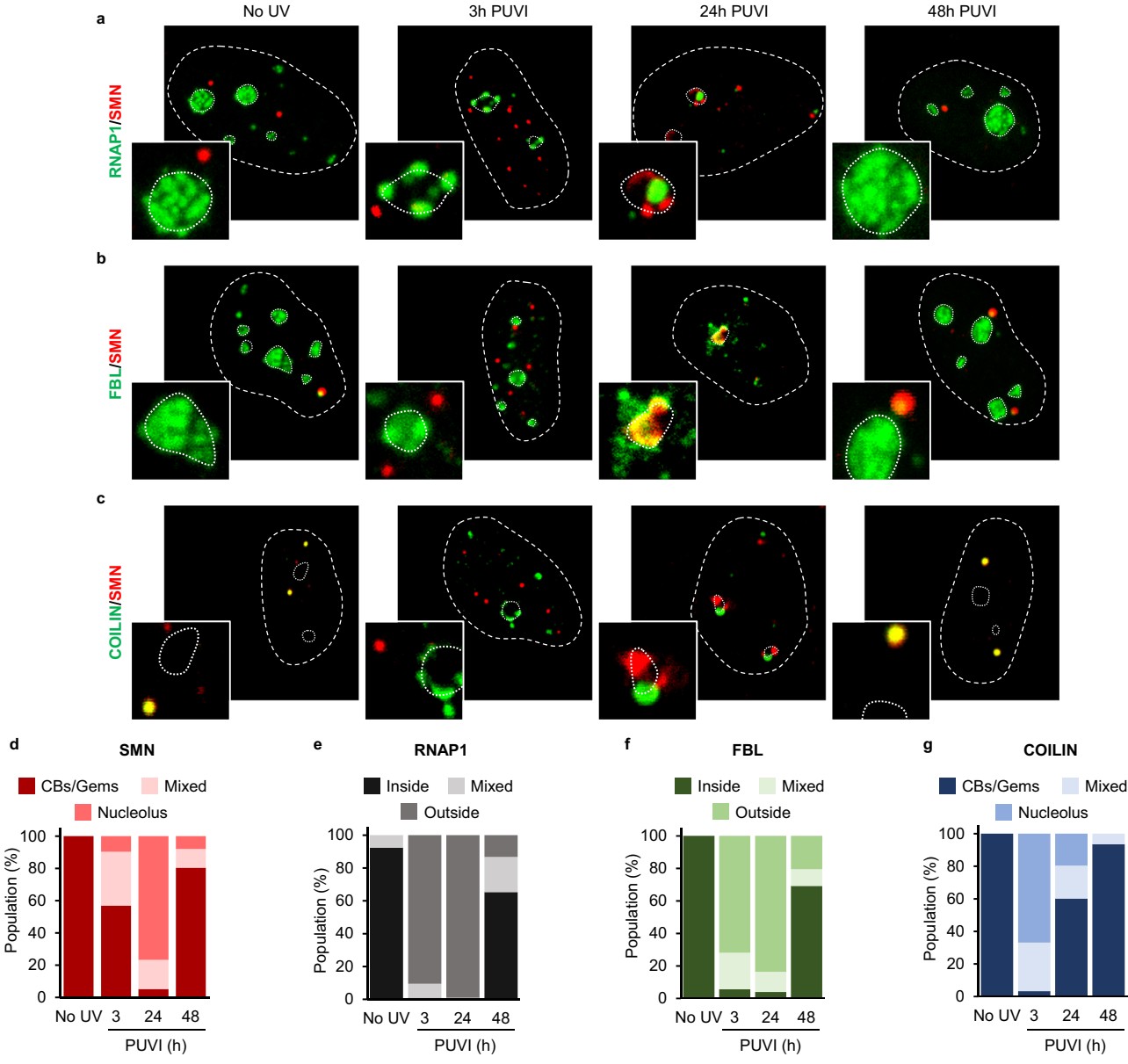

**Fig. 2 | Localization of SMN and its partners during DNA repair. a–c** Representative microscopy images of immunofluorescence (IF) assay in MRC5 cells showing, after 16 J/m² UV-C irradiation, the localization of SMN (red) and (**a**) RNAP1, (**b**) FBL or (**c**) Coilin (green) at different times Post UV-Irradiation (PUVI). Nuclei and nucleoli are indicated by dashed and dotted lines respectively. Scale bar: 5 μm. **d–g** Quantification of cells number for localization of (**d**) SMN, (**g**) Coilin (in Cajal Bodies [CBs] or Gems, at the periphery of the nucleolus or mixed localization), (**e**) RNAP1 and (**f**) FBL (inside the nucleolus, outside the nucleolus or mixed localization) at different times PUVI. For Coilin, RNAP1 and FBL, at least 50 cells from one representative experiment were analyzed. The SMN graph represent the average of three independent experiments, of which at least 50 cells for each were analyzed. Source data of graphs are provided as a Source Data file.

that FBL-SMN interactions require an intact TUDOR domain (Fig. S5e, f). We, therefore, examined the localization of SMN and FBL during the DNA repair process (Fig. 2b, quantification in 2 f). We observed a substantial co-localization between SMN and FBL at 24 h PUVI within the nucleolus (Fig. 2b, panel 24 h PUVI). The co-localization can also be observed at 40 h PUVI in the population of cells that, at this time point, have not yet restored the FBL position within the nucleolus (Fig. S5b).

In the absence of damage, SMN co-localizes and interacts with Coilin in CBs[15], however after DNA damage induction, it has been shown that CBs are disrupted[23]. Because of this evidence, we examined the localization of Coilin and its interactions with SMN during the DNA repair process. Coilin is localized to the periphery of the nucleolus already at 3 h PUVI (Fig. 2c, quantification in 2 g, panel 3 h PUVI) and remains in this location at 24 h PUVI (Fig. 2c, panel 24 h PUVI) and at 40 h PUVI (Fig. 2c, panel 40 h PUVI) in the subset of cells that did not yet reposition RNAP1. When Coilin is at the periphery of the nucleolus, no colocalization with SMN is observed (Fig. 2c).

To investigate whether the shuttling of SMN within the nucleolus is specific to UV damage or also happening when cells are submitted to other types of damage, we treated MRC5-SV cells with KBrO$_3$, known to produce a majority of 8-oxo-Guanine lesions[24] and examined the localization of SMN. When treated with KBrO$_3$, SMN shuttling into the nucleolus was also observed (Fig. S6a). Additionally, to investigate whether the SMN shuttling is a general event in stress response, we treated the MRC5-SV cells with Cordycepin, an RNAP1 transcription-blocking drug (Fig. S6b) and checked the localization of SMN (Fig. S6c). After cordycepin treatment (3 h), SMN shuttling was observed into the nucleolus but with a kinetic that was faster than the one measured after damage induction (Fig. S6c).

Finally, in order to visualize SMN shuttling in living cells, we produced a Cherry-SMN cell line by transfecting a Cherry-SMN expressing plasmid into the inducible sh6-SMN cell line and selecting a suitable clone that presented the correct SMN localization and level of expression (Fig. S7a). After depletion of the endogenous SMN, by induction of the shRNA against SMN with doxycycline, we UV-irradiated and performed time-lapse imaging over the following 48 h, with intervals of 1 h per image (Fig. S7b). During the time-lapse, we could detect the fading of the focal Cherry-SMN signal at around 3 to 6 h PUVI and the concomitant appearance of Cherry-SMN around the nucleoli. This signal lasted at least 24 h for some cells and disappeared at around 32 h PUVI with the reappearance of the focal pattern (CBs or Gems) around 40 h PUVI (Fig. S7b).

## SMN complex shuttles at the nucleolus after UV irradiation

In contrast to Coilin (Fig. 2c), at 3 h PUVI, SMN is still visible in a focal pattern within the nucleus, reminiscent of Gems. Because of this Coilin-independent localization and to investigate whether SMN shuttles within the nucleolus as an individual protein or as a complex, we investigated whether Gemin2, 3, 4 and 5 (subunits of the SMN complex) change location after UV-irradiation. We could show that Gemin 3, Gemin 4 and Gemin 5 interact with SMN all along this process of displacement and repositioning (Fig. 3a–c and corresponding quantifications in 3e, 3f and 3g), suggesting that the whole SMN complex is likely involved in the shuttling, or at least some components of the complex. It should be noted that Gemin5 is also present in foci that are SMN negative, these foci were visible all along the nucleolar reorganization process (Fig. 3c). Gemin 2 presented a different localization than SMN in both undamaged and damaged conditions. Gemin 2 was found to be present in both CBs/Gems and in the nucleolus in undamaged cells (Fig. 3d and corresponding quantification 3 h). After damage induction, the proportion of CBs/Gems containing Gemin2 decreased and concomitantly the nucleolar localization increased. To exclude that this nucleolar localization could be the result of a non-specific signal within the nucleolus, we quantified the Gemin2 signal within the nucleolus and observed that the signal

increased at 24 to 48 h PUVI (Fig. 3i). It is important to note that at 48 h PUVI, most Gemin 2 was still found within the nucleolus (Fig. 3h), while SMN was mainly localized in CBs/Gems (Fig. 2d). These results might indicate that some SMN partners follow exactly the same localization of SMN during damage-induced nucleolar reorganization, while others might be delayed in some steps of this process.

It has been previously shown that when SMN is reduced, Gemins levels are also reduced[25]. We have analysed by western blot the steady state levels of Gemin2, 3, 4, 5, 6, 7 and 8, in presence and absence of SMN. We determined by western blot that, Gemin2, 3, 4, 7, and 8 cellular concentration is SMN dependent, namely when SMN is depleted, the amount of these Gemins is decreased and when SMN concentration is rescued with a GFP-SMN WT expressing construct, the concentration of Gemins is rescued. Gemin6 antibody did not show any signal in WB and Gemin5 concentration is not modified in SMN knocked down cells (Fig. S8).

To complete this study, we have explored the possibility that Sm proteins would also shuttle at the nucleolus after DNA damage induction. In fact, previous work have detect some of the Sm proteins in proximity of the nucleolus[26]. Sm proteins interact with the SMN complex in the cytoplasm and are localised in nuclear speckles[27]. Indeed. without any DNA damage, Sm proteins are visualised in nuclear speckles, however, after DNA damage induction, Sm proteins accumulate at the periphery of the nucleolus with a similar initial kinetics than SMN, nevertheless, their return to a normal localisation (within nuclear speckles) is not yet completed at 48 h. We could demonstrate that this change in localisation is not SMN dependent, as no particular change was observed in shSMN cells (Fig. S9).

## SMN interacts with FBL, but not with coilin, in the nucleolus after UV irradiation

We showed that SMN co-localizes with FBL within the nucleolus at 24 h PUVI (Fig. 2b panel 24 h PUVI). To assess the in vitro interaction between SMN and FBL after UV-C exposure, we performed a GST pull-down assay using cell extracts untreated or UV-treated at different time points (Fig. S10a). We observed that UV treatment seemed to enhance FBL-SMN interactions, while GST alone failed to associate with FBL even if more GST alone than GST-SMN was used in pull-down assays. To confirm this result in vivo, we performed a Proximity Ligation Assay (PLA) on wild-type cells at different times PUVI (Fig. S10b, c). The majority of the cells at 24 h PUVI presented a strong PLA signal specifically in the nucleolus between SMN and FBL (Fig. S10c) which could be quantified as a specific interaction signal (Fig. S10b).

No colocalization was found between SMN and Coilin during the shuttling process, neither at 3 h nor later on (Fig. 2c). SMN and Coilin did colocalize in CBs before damage induction and 48 h post damage induction. To confirm this result, we performed PLA on wild-type cells at different times PUVI (Fig. S10d, e) and we could not detect any added interaction/proximity between SMN and Coilin during damage-induced nucleolar reorganization.

## Coilin is required for SMN import into the nucleolus and for the nucleolar rearrangement following UV-C exposure

We observed that Coilin localized to the nucleolus during UV damage (at 3 h PUVI) prior to SMN (Fig. 2c, g). These results led us to hypothesize that Coilin might be the factor that recruits SMN to the nucleolus. To test this idea, we depleted cells of Coilin by using a pool of four specific siRNAs and two individual siRNAs (Fig. S11a) and performed IF of SMN and RNAP1 on wild-type cells before damage and at different time PUVI (Fig. 4a) in the presence or absence of Coilin. What could be noticed was that the number of cells presenting SMN foci was diminished when Coilin was downregulated (Fig. S11b), this result is consistent with the fact that CBs are dismantled in the absence of Coilin. This would mean that, in this condition, visible SMN foci would more likely correspond to Gems. To investigate whether Gems would

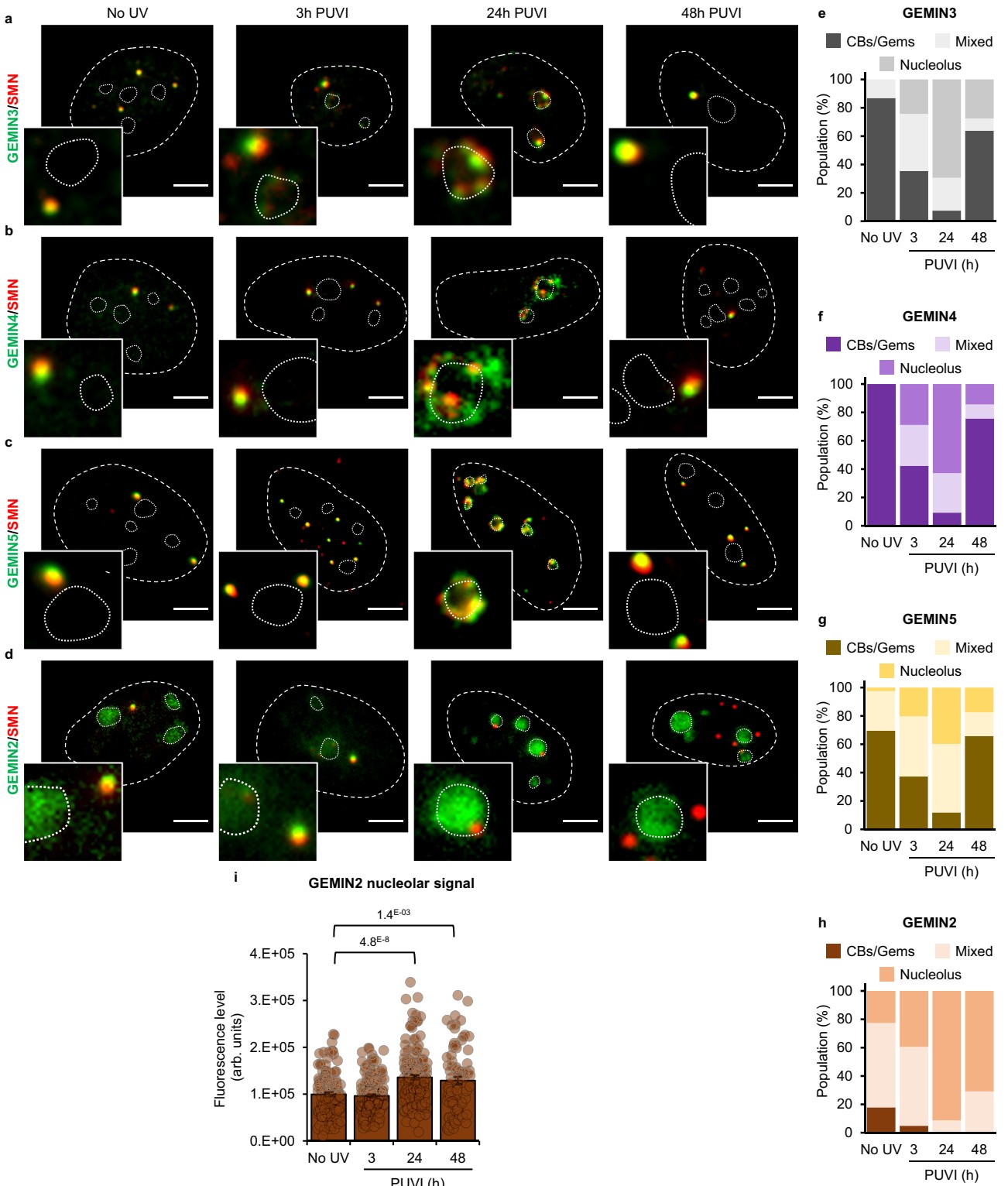

**Fig. 3 | Localization of Gemin2, Gemin3, Gemin4 and Gemin5 during DNA repair. a–d** Representative microscopy images of immunofluorescence (IF) assay in MRC5 cells showing, after 16 J/m² UV-C irradiation, the localization of SMN (red) and (**a**) GEMIN3, (**b**) GEMIN3, (**c**) GEMIN4 or (**d**) GEMIN2 (green) at different times Post UV-Irradiation (PUVI). Nuclei and nucleoli are indicated by dashed and dotted lines respectively. Scale bar represents 5 μm. **e–h** Quantification of cells number for localization of (**e**) GEMIN3, (**f**) GEMIN4, (**g**) GEMIN5 or (**h**) GEMIN2 (in Cajal Bodies [CBs] or Gems, at the periphery of the nucleolus or mixed localization), at different times PUVI. At least 50 cells from one representative experiment were analyzed. **i** Quantification of fluorescent signal in the nucleolus from the IF against GEMIN2. Data are represented as mean values +/− SEM. At least 90 cells was quantified from one representative experiment. The *p*-value correspond to a student's test with two-tailed distribution and two-sample unequal variance to compare after irradiation with No UV condition. Source data of graphs are provided as a Source Data file.

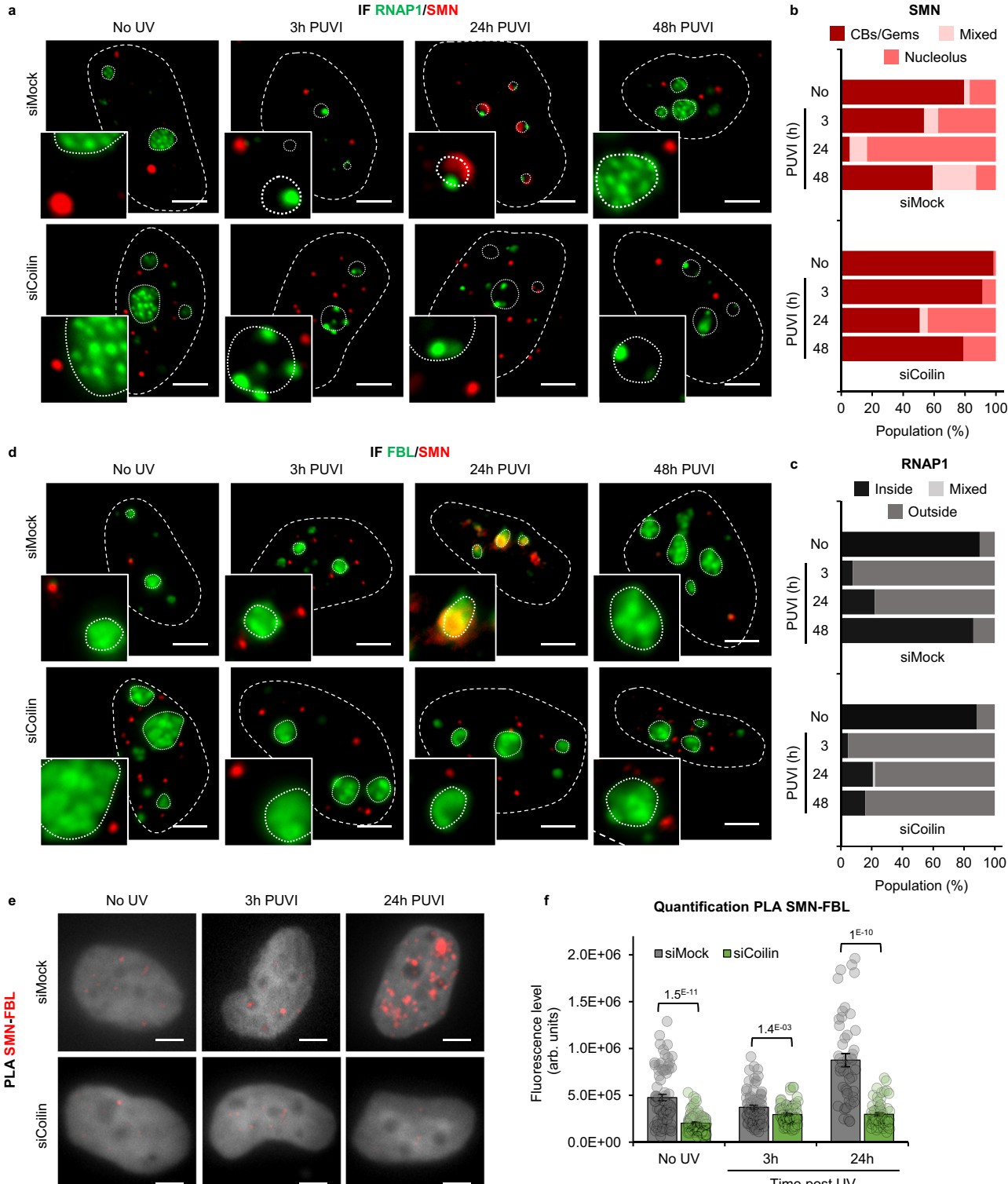

**Fig. 4 | SMN shuttling is Coilin-dependent. a, d** Representative microscopy images of immunofluorescence (IF) assay showing the localization of SMN (red) and (**a**) RNAP1 or (**d**) FBL (green) at different times Post UV-Irradiation (PUVI) in cells transfected with siMock or siCoilin pool. Nuclei and nucleoli are indicated by dashed and dotted lines respectively. Scale bar: 5 µm. **b, c** Quantification of cells number for localization of (**b**) SMN (in Cajal Bodies [CBs] or Gems, at the periphery of the nucleolus or mixed localization) and (**c**) RNAP1 (inside the nucleolus, outside the nucleolus or mixed localization) at different times PUVI in cells transfected with siMock or siCoilin pool. At least 35 cells from one representative experiment were analyzed. **e** Representative microscopy images of Proximity Ligation Assay (PLA) in red showing the interaction between SMN and FBL at different times PUVI in cells transfected with siMock or siCoilin pool. Scale bar: 5 µm. DAPI in grey. **f** Quantification of fluorescent signal in the nucleus against the couple SMN-FBL from PLA experiment in cells transfected with siMock or siCoilin pool after UV-C irradiation. Data are represented as mean values +/− SEM. At least 45 cells was quantified from one representative experiment. The *p*-value correspond to a student's test with two-tailed distribution and two-sample unequal variance to compare siCoilin with siMock. Source data of graphs are provided as a Source Data file.

shuttle to the nucleolus, we excluded from the quantification cells that did not present SMN foci. Our results show that, without Coilin, the shuttling of SMN within the nucleolus is impaired and SMN remains mostly in Gems (Fig. 4a and corresponding quantification 4b and S11c). Consequently, in cells deficient for Coilin, the nucleolar reorganization after DNA damage and repair is not restored and RNAP1 remains at the periphery of the nucleolus at 48 h PUVI (Figs. 4a, c and S11d), indicating that the shuttling of SMN and the presence of Coilin are both important to ensure the re-establishment of the nucleolar reorganization.

To investigate whether the absence of Coilin would impact the interaction between SMN and FBL, we performed PLA and IF of SMN and FBL on wild-type cells before damage and at different times PUVI in the presence or absence of Coilin (Fig. 4d–f). Our results show that without Coilin, SMN remains outside of the nucleolus and no co-localization with FBL was observed (Fig. 4d). As a consequence, the interaction between SMN and FBL observed at 24 h PUVI (Fig. S10b, c) is lost in Coilin-depleted cells (Fig. 4e, f).

### FBL is required for SMN export from the nucleolus and for the nucleolar rearrangement following UV-C exposure

As Coilin, FBL is an essential partner of SMN and a nucleolar protein that can methylate rDNAs and Histones within the nucleolus[21,28]. We showed that FBL and SMN interact in cell extracts by GST (Figs. S5e, S10a) and in vivo by PLA (Fig. S10b, c), notably stronger after SMN shuttling into the nucleolus (24 h PUVI). As the SMN shuttling is dependent on Coilin (Fig. 4), we wondered whether FBL depletion would also play a role in this shuttling process. To test this hypothesis, we depleted cells of FBL by using a specific siRNA pool (Fig. S12a) or 2 independent siRNAs (Fig. S12b) and performed IF of SMN and RNAP1 on wild-type cells before damage and at different times PUVI (Fig. 5a, corresponding quantifications in Fig. 5b, c) in presence or absence of FBL. Our results show that without FBL, the shuttling of SMN within the nucleolus is altered, namely SMN localizes at the periphery of the nucleolus already at 3 h PUVI and stays at the periphery of the nucleolus at all time points (Figs. 5a, b and S12c). In cells deficient for FBL, the nucleolar reorganization after DNA damage and repair is not restored and RNAP1 remains at the periphery of the nucleolus at 48 h PUVI (Figs. 5a, c and S12d), indicating that the shuttling of SMN within the nucleolus and the presence of FBL are both important to ensure the re-establishment of the nucleolar reorganization.

To verify if and how the absence of FBL affects the localization and interaction between SMN and Coilin, we performed IF and PLA of SMN and Coilin on wild-type cells before damage and at different times PUVI (Fig. 5d–f) in the presence or absence of FBL. Our results show that, without FBL, the interaction between SMN and Coilin is detectable at all times PUVI (Fig. 5e, f) and both SMN and Coilin are localized at the periphery of the nucleolus already at 3 h PUVI, this localization does not change at 24 h or 40 h PUVI (Fig. 5d). These findings show that FBL is also a critical player in SMN shuttling and in nucleolar reorganization following UV irradiation and DNA repair.

### SMN mutants are deficient in nucleolar reorganization after DNA damage induction and repair

SMN protein has different domains that are important for its functionality, interactions with partners and stability (Fig. S5f). In order to study how different SMN domains impact the nucleolar reorganization function of SMN, we have focused on two SMN mutants: (i) a mutant in the Tudor domain (E134K) and (ii) a mutant in the YG domain (Y272C). The Tudor domain is important for the interaction of SMN with FBL and Coilin[21], while the YG domain is essential for the oligomerization of SMN[29] which is more stable than the monomeric form.

We investigated whether these mutated SMN might have a defect in nucleolar reorganization after DNA damage induction or in the shuttling of SMN to and/or from the nucleolus. We produced three GFP-SMN cell lines by transfecting into the inducible sh6-SMN cell line

(i) a wild-type GFP-tagged SMN (hereafter GFP-SMN); (ii) a mutant GFP-tagged SMN, bearing the mutation in the Tudor domain E134K (hereafter GFP-SMN^E134K); (iii) a mutant GFP-tagged SMN, bearing the mutation in the YG domain Y272C (hereafter GFP-SMN^Y272C). After the selection of suitable clones, we verified the amount of GFP-SMN and the mutant SMN proteins produced and we could determine that the amount of mutant SMN was lower than the GFP-SMN protein (Fig. S13a). After depletion of the endogenous SMN, by induction of the sh6-SMN RNA, we UV-irradiated SMN wild type and mutant expressing clones and we performed IF against RNAP1 coupled to the direct visualization of the GFP-SMN, GFP-SMN^E134K or GFP-SMN^Y272C signal. Wild-type and mutant SMN proteins could be visualized in CBs/Gems in undamaged cells (Fig. S13b, c). Nevertheless, the Tudor mutant GFP-SMN^E134K expressing cells failed to restore CBs/Gems at 48 h PUVI and the nucleolar reorganization was not restored, RNAP1 remaining at the periphery of the nucleolus at 48 h PUVI (Fig. S13b, c, d). These results are reminiscent of the results observed in the absence of FBL (Fig. 5a–c). The YG mutant GFP-SMN^Y272C failed to properly shuttle to the nucleolus and remained mostly in CBs/Gems all along the nucleolar reorganization process (Fig. S13b, c) and the nucleolar reorganization was not restored at 48 h PUVI (Fig. S13b, d). These results are reminiscent of the results observed in the absence of COILIN (Fig. 4a–c). To resume, both SMN mutants studied, show that a functional SMN is needed to properly ensure nucleolar homeostasis after DNA damage induction and that in different ways SMN shuttling is perturbed by these mutations.

### PRMT1 activity mediates the nucleolar shuttling of SMN

One of the activities of SMN is to bind, via the Tudor domain, Arginine methylated proteins[30,31]. Arginine methylation is a widespread post-translational modification that can occur in histones and non-histone proteins[32–35]. The enzymes catalyzing the transfer of a methyl group to Arginine residues are part of a family called the PRMTs (Protein Arginine Methyl Transferases). PRMTs can mono-methylate Arginine residues (MMA) or di-methylate Arginine residues either symmetrically (SDMA) or asymmetrically (ADMA). Because these proteins affect SMN functions but also the interaction of SMN with Coilin[36], we wondered whether one of the PRMTs could affect, disturb or enhance SMN shuttling after DNA damage. Using GST pull-down assay, we observed that PRMT1 interacts with SMN (Fig. S14a). We thus decided to deplete cells from PRMT1 by siRNA silencing (Fig. S14b) and performed IF against SMN and FBL. We found that, without PRMT1, the shuttling of SMN to the nucleolus is inhibited (Fig. 6a, b). In fact, in PRMT1-depleted cells, SMN remains mostly in CBs/Gems. This SMN localization after DNA damage is reminiscent of the one observed in Coilin-depleted cells (Fig. 4a).

To verify whether the perturbation of SMN shuttling observed in Fig. 6a is due to the physical depletion of PRMT1 or the inhibition of the ADMA methylase activity, we treated the cells with the PRMT-class I specific inhibitor MS023[37] and the more specific and potent PRMT1 inhibitor, Furamidine[38] (Figs. 6d, e, S14c, d) prior to DNA damage and IF assays. We could verify that the inhibition of the methylase activity of PRMTs from class I and specifically of PRMT1 perturbs SMN shuttling (Figs. 6d, e and S14c) and that SMN remains mostly in CBs/Gems as when PRMT1 is depleted (Fig. 6a). Because the SMN complex requires the protein arginine methyltransferase 5 (PRMT5) to assemble Sm core structures of spliceosomal U-rich small nuclear ribonucleoprotein particles[39], we investigated whether depletion of PRMT5 would affect SMN shuttling to the nucleolus. We down-regulated PRMT5 by siRNA (Fig. S15d), induced DNA damage and perform IF against SMN and RNAP1 at different times PUVI (Fig. S15a). Our results show that depletion of PRMT5 does not influence SMN shuttling (Fig. S15a, b), nor the nucleolar homeostasis after DNA damage and repair (Fig. S15a, c).

To investigate how the depletion of PRMT1 affects the interactions between SMN and its partners (Coilin and FBL), we performed

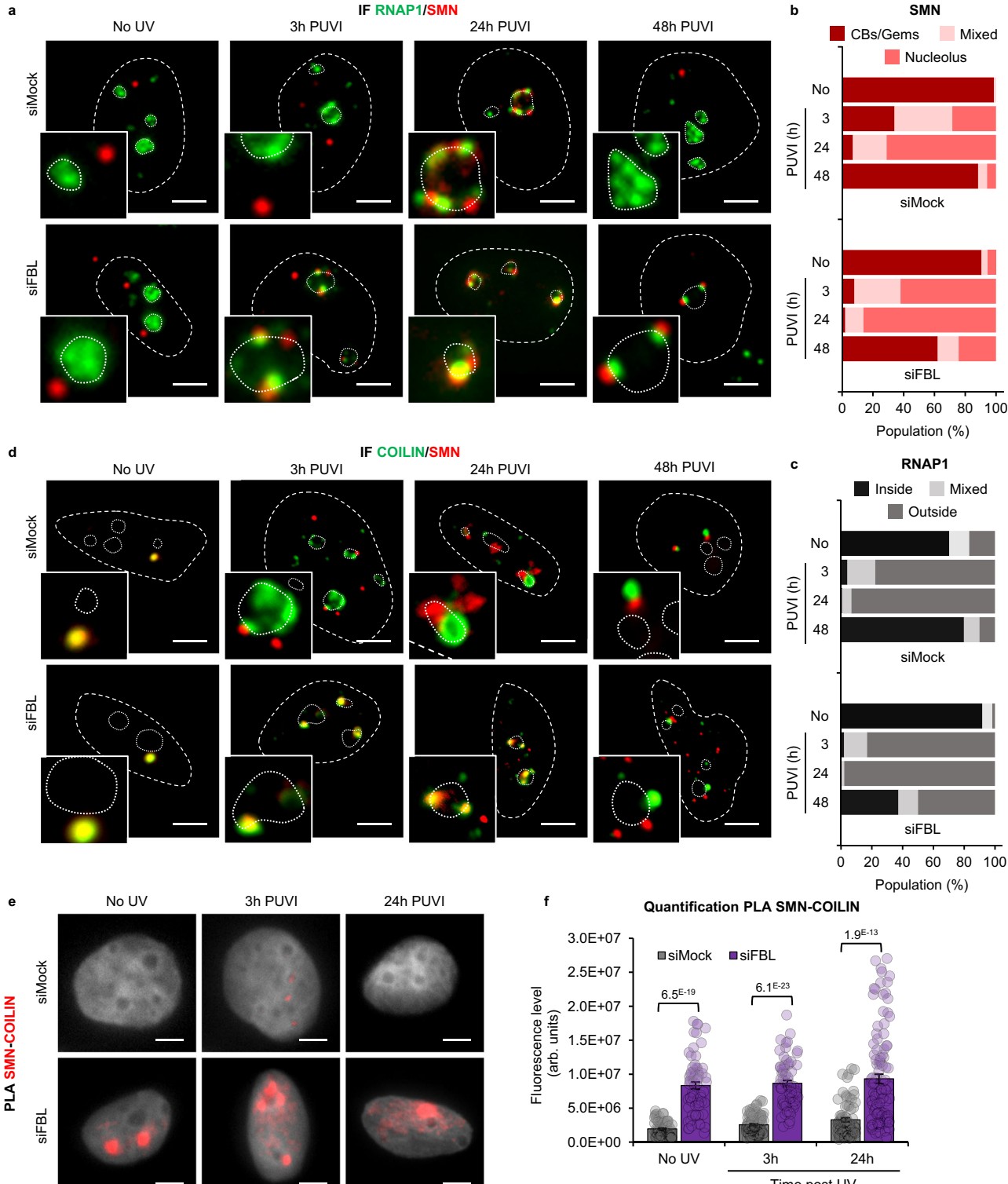

**Fig. 5 | The release of SMN from the nucleolus is FBL-dependent.**
**a**, **d** Representative microscopy images of immunofluorescence (IF) assay showing the localization of SMN (red) and (**a**) RNAP1 or (**d**) COILIN (green) at different times Post UV-Irradiation (PUVI) in cells transfected with siMock or siFBL pool. Nuclei and nucleoli are indicated by dashed and dotted lines respectively. Scale bar: 5 μm. **b**, **c** Quantification of cells number for localization of (**b**) SMN (in Cajal Bodies [CBs] or Gems, at the periphery of the nucleolus or mixed localization) and (**c**) RNAP1 (inside the nucleolus, outside the nucleolus or mixed localization) at different times PUVI in cells transfected with siMock or siFBL pool. At least 80 cells from one representative experiment were analyzed. **e** Representative microscopy images of

Proximity Ligation Assay (PLA) in red showing the interaction between SMN and COILIN at different times PUVI in cells transfected with siMock or siFBL pool. Scale bar: 5 μm. DAPI in grey. **f** Quantification of fluorescent signal in the nucleus against the couple SMN-COILIN from PLA experiment in cells transfected with siMock or siFBL pool after UV-C irradiation. Data are represented as mean values +/− SEM. At least 65 cells was quantified from one representative experiment. The p-value correspond to a student's test with two-tailed distribution and two-sample unequal variance to compare siFBL with siMock. Source data of graphs are provided as a Source Data file.

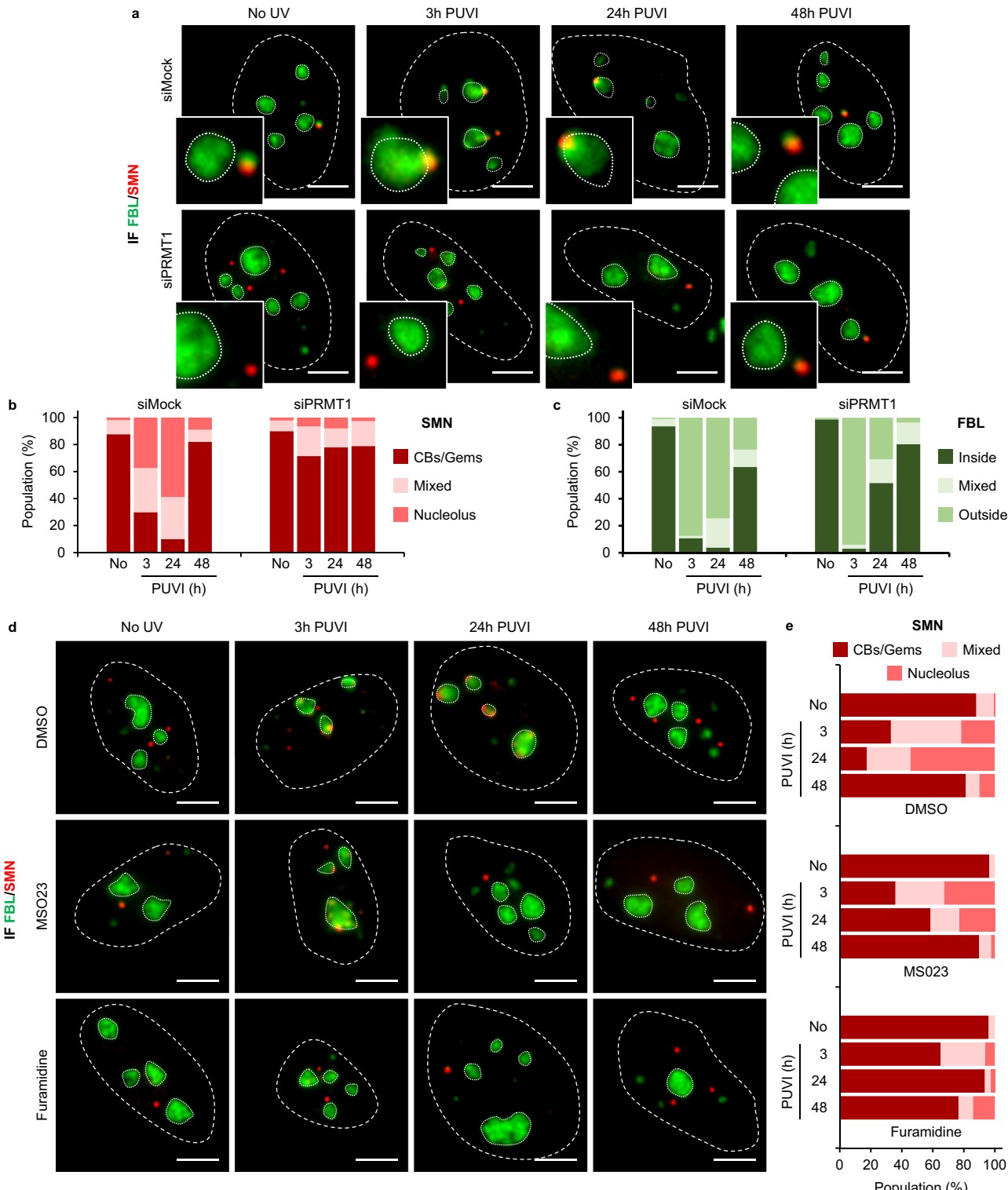

**Fig. 6 | PRMT1 activity mediates the nucleolar shuttling of SMN.**
**a** Representative microscopy images of immunofluorescence (IF) assay showing the localization of SMN (red) and FBL (green) at different times Post UV-Irradiation (PUVI) in cells transfected with siMock or siPRMT1 pool. Nuclei and nucleoli are indicated by dashed and dotted lines respectively. Scale bar: 5 μm. **b, c** Quanti®ca-tion of cells number for localization of (**b**) SMN (in Cajal Bodies [CBs] or Gems, at the periphery of the nucleolus or mixed localization) and (**c**) FBL (inside the nucleolus, outside the nucleolus or mixed localization) at different times PUVI in cells transfected with siMock or siPRMT1 pool. At least 145 cells from one

representative experiment were analyzed. **d** Representative microscopy images of IF assay showing the localization of SMN (red) and FBL (green) in MRC5 cells treated with DMSO, MS023 or Furamidine followed by 16 J/m² UV-C irradiation. Nuclei and nucleoli are indicated by dashed and dotted lines respectively. Scale bar: 5 μm. **e** Quantification of cells number for localization of SMN (in Cajal Bodies [CBs] or Gems, at the periphery of the nucleolus or mixed localization) at different times PUVI in cells treated with DMSO, MS023 or Furamidine. At least 60 cells from one representative experiment were analyzed. Source data of graphs are provided as a Source Data file.

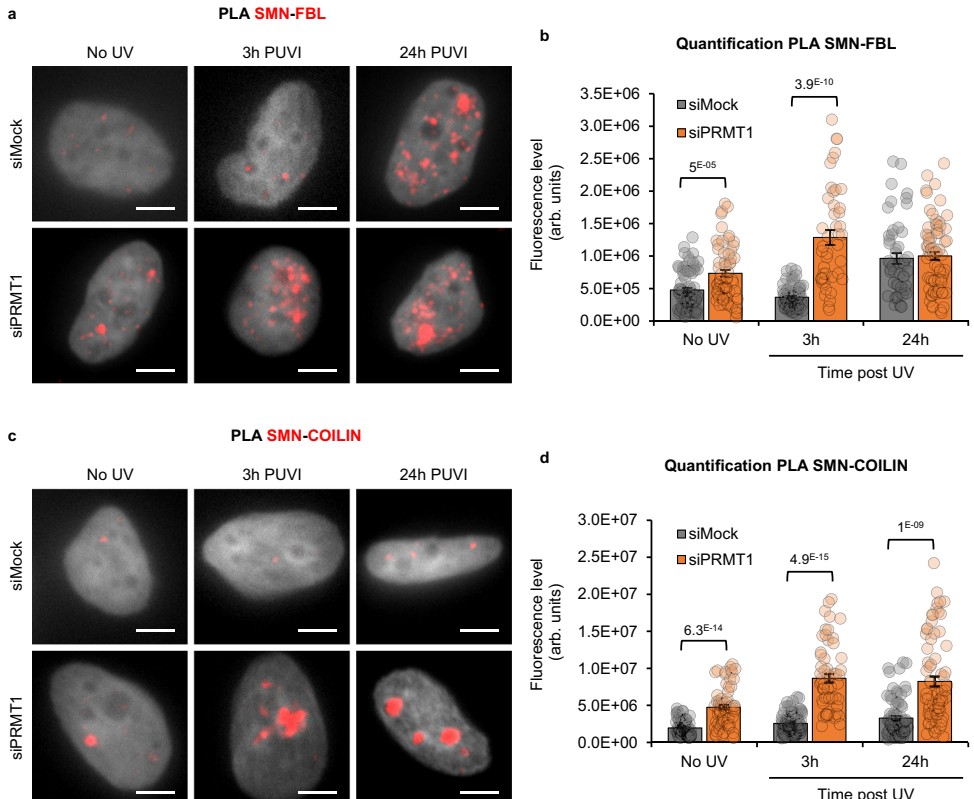

**Fig. 7 | PRMT1 remodels the interaction of SMN with FBL and COILIN.**
**a, c** Representative microscopy images of Proximity Ligation Assay (PLA) in red showing the interaction (**a**) between SMN and FBL or (**c**) between SMN and COILIN at different times Post UV-Irradiation (PUVI) in cells transfected with siMock or siPRMT1 pool. Scale bar: 5 μm. DAPI in grey. **b, d** Quantification of fluorescent signal in the nucleus against the couple (**b**) SMN-FBL or (**d**) SMN-COILIN from PLA experiment in cells transfected with siMock or siPRMT1 pool after UV-C irradiation. Data are represented as mean values +/− SEM. At least 45 cells was quantified from one representative experiment. The *p*-value correspond to a student's test with two-tailed distribution and two-sample unequal variance to compare siFBL with siMock. Source data of graphs are provided as a Source Data file.

PLA assays and measured a stronger interaction of SMN and FBL before damage induction and at 3 h PUVI which corresponds with the beginning of SMN shuttling at the nucleolus (Fig. 7a, b). We also quantified a stronger interaction of SMN with Coilin when PRMT1 is depleted (Fig. 7c, d) before damage induction and up to 48 h PUVI.

We showed by IF that PRMT1 is also shuttling during nucleolar reorganization in wild-type cells and could be detected at 6 h and up to 48 h PUVI within the nucleolus or at the periphery of the nucleolus (Fig. S16a, b). The shuttling of PRMT1 at the nucleolus is dependent on the SMN protein as in SMN-depleted cells (Sh6-SMN), PRMT1 is not detected inside the nucleoli but remains mainly nuclear or localized at the periphery of the nucleolus (Fig. S16a, b).

We could quantify an increased level of PRMT1 after damage induction and we could show that this increase is partially dependent on the presence of a functional SMN (Figs. S16c, S17a, b).

**SMN cells are sensitive to DNA damage**
We demonstrated that a functional SMN is important for proper nucleolar homeostasis after DNA damage repair and wondered whether this newly discovered function could have a biological consequence on SMN cells and in fine on SMA patients. In order to investigate whether DNA damage could impact the survival of SMN cells we performed clonogenic assays on sh6-SMN cells with and without induction of the sh expression and compared the results with a DNA repair-deficient cell lines (CS1AN-SV) mutated in the CSB protein, involved in transcription coupled-NER (Fig. 8a). Our results show that SMN-depleted cells are moderately sensitive to UV damage as their sensitivity to UV irradiation is intermediate between an SMN wild-type cell line and a DNA repair-deficient cell line.

Because we demonstrated that SMN shuttling is also happening during oxidative damage repair (Fig. S6a), we wondered whether SMN cells would be sensitive to chronic oxidative damage. We usually culture rodents, human patients or mutated cell lines in physioxia conditions (3% of oxygen) to avoid the supraphysiological oxygen levels disturbing normal cellular metabolism and increasing the cellular concentration of reactive oxygen species[40]. To induce chronic oxidative damage in SMN-depleted cells, we culture them in a standard incubator at 20% oxygen and compare their ability to form colonies. As with many established cell lines, Human SV40-immortalized fibroblasts MRC5 (MRC5-SV) have a better clonogenicity in physioxia conditions (Fig. 8b). However, MRC5-SV expressing shRNA SMN has a decreased clonogenicity compared to MRC5-SV that expresses a mock shRNA (Fig. 8b), which demonstrates that the absence of SMN per se hinders the ability of cells to form clones. When SMN-depleted cells are grown in supraphysiological oxygen levels their clonogenicity is reduced, compared with the physioxia culture conditions (Fig. 8c).

## Discussion

One of the least explored areas in the DNA repair field is how cells restore their cellular activities after the completion of all the reactions that allow cells to eliminate DNA lesions. Most DNA lesions block transcription and replication[5] and although we have extensive knowledge of how cells recognize and repair DNA lesions, very little is known about how cells restart these cellular processes. In post-mitotic cells, restoration of the damage-induced block of transcription is essential for cell survival.

We have previously shown that RNAP1 transcription is blocked after UV lesions and that TC-NER pathway is responsible for the

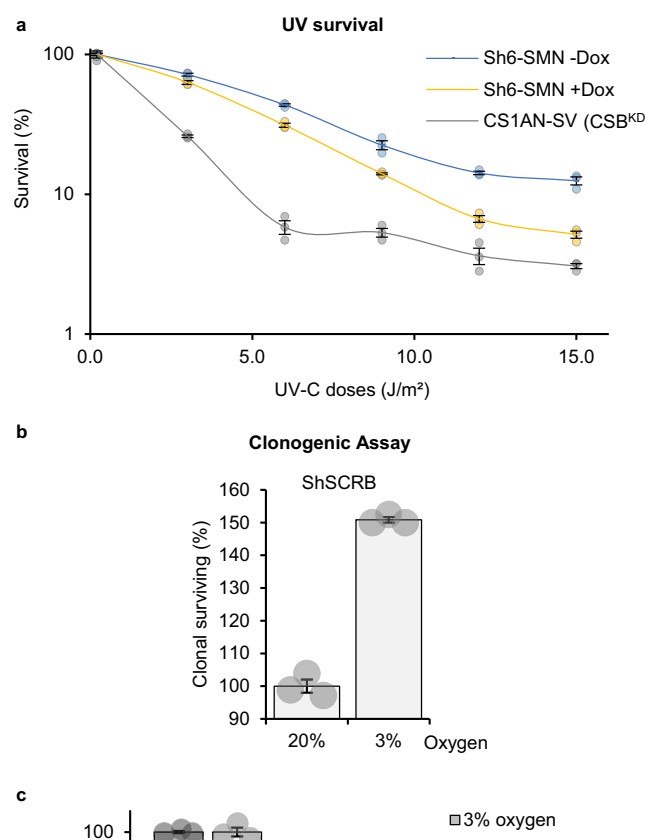

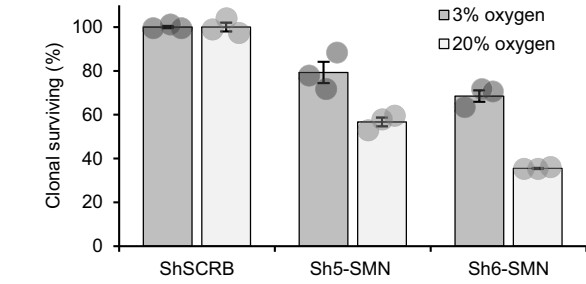

**Fig. 8 | SMN-deficient cells are sensitive to DNA damage. a** Survival curve was determined by the colony-forming ability to UV-C of cells expressing (+Dox) or not (−Dox) Sh6-SMN. Counting of clones was normalized to non-irradiated condition. **b, c** Clonogenic assay of cells cultured with different quantities of oxygen (3% or 20%) in the presence (ShSCRB = ShSCRAMBLE) or absence of SMN (Sh5-SMN and Sh6-SMN). Counting of clones was normalized (**a**) to 20% oxygen or (**b**) to ShSCRB. Data are represented as mean values of the triplicate +/− SEM. Source data of graphs are provided as a Source Data file.

repair of UV-lesions on ribosomal DNA[8]. UV-damages impact the organization of the nucleolus and during DNA repair both nucleolar DNA and RNAP1 are displaced at the periphery of the nucleolus[8]. Although RNAP1 transcription restarts when UV-lesions on the transcribed strand are repaired, the return of the RNAP1 within the nucleolus is dependent on the proper repair of DNA lesions on the untranscribed nucleolar DNA[8]. In this particular case (GG-NER deficient cells), RNAP1 transcription restarts in a non-canonical compartment, the periphery of the nucleolus, and this anomaly might influence the proper ribosome biogenesis. Therefore, restoring the proper nucleolar structure and organization might be important for cellular viability or the efficiency of cellular processes. The recovery of a normal nucleolar structure after DNA damage repair is not a passive process and requires the presence of some key proteins[9], although their exact mechanistic role has not been established yet.

In a quest to find the exact molecular mechanism for the re-establishment of the nucleolar organization after DNA repair completion, we set up a best-candidate approach that guided us to inspect

the consequences of different nucleolar proteins' interactors depletion. One of these candidates was the protein SMN, which is a particularly interesting protein to scrutinize because of the known interaction with the nucleolar protein FBL via its Tudor domain[17].

SMN is part of a multiprotein complex that includes Gemins2-8 and Unrip[41] and acts like a molecular chaperone needed to assemble small ribonucleoproteins (snRNPs) which are part of the spliceosome[42] and small nucleolar ribonucleoprotein particles (snoRNPs) involved in post-transcriptional processing of ribosomal RNA[21]. Additionally, SMN has been also found implicated in transcription[43], histone mRNA processing[44], mRNAs trafficking[45] and translation[46].

We have shown here that SMN plays a role in nucleolar homeostasis after DNA repair completion. In fact, silencing or mutations (Tudor domain and oligomerization mutants) of the SMN protein hinder the proper return of RNAP1 within the nucleolus (Fig. 9), despite the restart of RNAP1 transcription (Figs. 1, S1, S2, S3 and S13). Contrary to what has been observed in GG-NER deficient cells, the RNAP1 mislocalization after DNA repair completion is not induced by the presence of unrepaired DNA lesions because we could show that SMN deficient cells are proficient in the NER pathway, repairing UV-lesions on both transcribed and untranscribed regions of the genome (Fig. S4).

The multifunctional SMN is localized in both the cytoplasm and in nuclear bodies (Cajal Bodies and Gems) and in addition, SMN can also be detected in the nucleolus of neurons[16] although the function of SMN in nucleolus remains enigmatic. During the process of nucleolar reorganization, we observed a shuttling of SMN from CBs/Gems to the periphery of the nucleolus or within the nucleolus during RNAP1 displacement and a return to CBs/Gems after the RNAP1 repositioning (Figs. 2, 9). SMN shuttles to the nucleolus together with its partners Gemin 2, Gemin 3, Gemin 4 and Gemin 5 (Fig. 3). While Gemin 3 and 4 followed the same SMN localization, the classical SMN partner Gemin 2[47] was observed to have both CBs/Gems and nucleolar localization. While SMN/Gemin2 foci disappeared during RNAP1 displacement, the nucleolar signal of Gemin 2 increased reaching the maximum after RNAP1 repositioning and Gemin2 foci did not reappear. It is still unknown what retains Gemin2 within the nucleolus at late time points and this retention might influence the reconstitution of nuclear SMN complex. Notably, Gemin 5 was observed in both SMN positive and negative foci which could be because a large fraction of Gemin5 protein is found outside of the SMN complex[48]. SMN negative foci of Gemin5 were present at any time point throughout the nucleolar reorganization process, which might indicate that SMN negative foci of Gemin5 do not participate in the shuttling of SMN complex and in nucleolar homeostasis.

SMN shuttling happens in replicative and post-mitotic cells (motoneurons) and in different stress situations: UV-induced DNA damage, RNAP1 transcriptional block and drug-induced oxidative damage. Notably, we should highlight that the kinetics of SMN shuttling might vary depending on different types of stresses but also on the culture conditions, the amount and quality of serum, supraphysiological oxygen levels and the type of culture media. The kinetic of SMN shuttling should be defined for each type of stress/damage and culture condition.

SMN shuttling is dependent on both Coilin and FBL (Figs. 4, 5, 9). In Coilin-depleted undamaged cells, we observed a high number of cells in which no CBs or other foci were visible, this might be because, without Coilin, CBs are disrupted and just Gems are visible. In the absence of Coilin, we observe an increased number of Gems and the number of cells in which SMN is at the nucleolus is reduced. In Coilin-depleted cells, RNAP1 repositioning is hindered, indicating that in the absence of Coilin nucleolar reorganization is compromised. In FBL-depleted cells, SMN remains at the periphery of the nucleolus at all time points and RNAP1 does not recover the proper localization within the nucleolus but remains at the periphery of the nucleolus.

Because FBL interacts with SMN via its Tudor domain[17] and SMN interacts mainly with methylated arginine residues, we explored

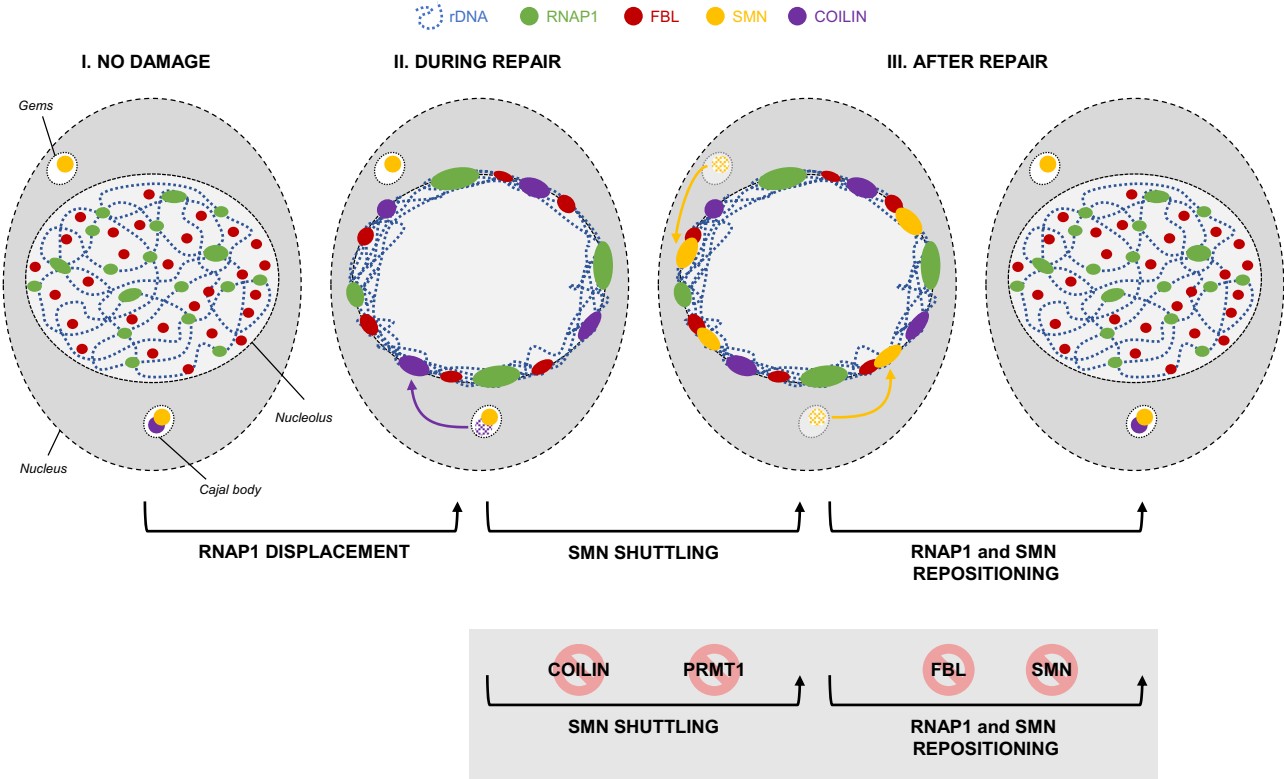

**Fig. 9 | SMN shuttling during DNA damage/repair dependent nucleolar reorganization.** After genotoxic stress, RNA Polymerase 1 (RNAP1 in green), Fibrillarin (FBL in red) and nucleolar DNA (rDNA in blue) are exported to the periphery of the nucleolus. During DNA repair, Coilin (in purple) and subsequently SMN (in yellow) shuttle from Cajal bodies to the nucleolus. Once DNA repair is fully completed, the organization of the nucleolus is restored. The shuttling of SMN relies on the presence of Coilin and the activity of PRMT1. The restoration of the nucleolar structure is dependent on FBL and SMN.

the possibility that one of the PRMTs would be responsible for SMN shuttling within the nucleolus. Indeed, the activity of PRMT1, responsible for ADMA, is essential to recruit SMN to the nucleolus (Figs. 6, 9). In contrast, depletion of PRMT5, which is responsible for the assembly of Sm proteins to the SMN complex did not influence the shuttling of SMN, probably indicating that the SMN function in nucleolar homeostasis is independent of the snRNPs biogenesis process[49] (Fig. S15).

Depletion of Coilin, FBL and PRMT1 also modifies the interactions between SMN, Coilin and FBL, by increasing their proximities (Figs. 5, 6, 7) which might indicate that specific methylations are important to maintain an equilibrium of interactions between these three partners.

PRMT1 shuttles within the nucleolus after DNA damage induction as SMN (Fig. 7). This shuttling within the nucleolus is SMN-dependent. Although we do not know if the substrate of PRMT1 is a protein that directly interacts with SMN, we showed that methylation is essential for SMN shuttling. It is plausible to assume that because FBL is methylated by PRMT1[50], FBL might be the substrate methylated during the process of RNAP1 repositioning within the nucleolus after the completion of DNA repair.

In the future years, it will be important to unveil the precise mechanism of action of SMN within the nucleolus, by investigating what are the partners of SMN in this organelle after DNA damage and how they function together with SMN, how CBs/Gems disassemble and reassemble, how they restore their activity and what are the consequences on splicing, ribosome biogenesis and finally translation fidelity. More generally, it will be important to continue to study how cells maintain and/or restore nucleolar homeostasis after cellular stresses, such as DNA damage.

As for other multifunctional proteins, it is difficult to identify exactly which function of SMN is responsible for the phenotype observed in SMA patients. Likely, the SMA phenotype could be due to a combination of different defects that would synergistically impact cellular activities. Our results present here a newly discovered SMN function in maintaining nucleolar homeostasis after DNA damage and repair. We showed that SMN-depleted cells show a sensitivity to UV-induced damage and most importantly to chronic oxidative damage (Fig. 8) which is what motoneurons would have to deal with all along patients' life. This results point to the fact that not just the DNA repair process is important for cell viability but also all the processes that restore cellular functions after DNA repair completion.

Our findings may directly impact the life and well-being of SMA patients[51]. In fact, in cells and motoneurons of SMA patients, exogenous and endogenous DNA damage might progressively and lastingly disrupt the nucleolar structure and disturb ribosome biogenesis leading to perturbed protein translation[52]. This defect may contribute to the neurodegenerative phenotype of SMA motoneurons. If this hypothesis is true, SMA patients could be advised to prevent deleterious DNA damage to avoid a reorganization of the nucleolus that would affect proper protein production within neurons. Although exogenous DNA damage is partially avoidable, endogenous DNA damage is inevitable and one way to reduce its overload is to follow a diet rich in antioxidants. This nutritional approach combined with a healthy lifestyle, avoiding exogenous damage, such as cigarette smoke, pollutants, and potentially harmful molecules, may retard the degeneration of motoneurons and thus SMA progression.

## Methods
### Cells culture
The cells used in this study for the main figure come from Erasmus MC in Rotterdam and were SV40-immortalized human fibroblasts, wild-type (MRC5-SV [RRID:CVCL_D690]) and CSB-deficient (CS1AN-SV, TC-

NER deficient [RRID:CVCL_L472]). They were cultured in DMEM supplemented with 10% fetal bovine serum (FBS) and 1% penicillin and streptomycin (P/S) and incubated at 37 °C with 3% or 20% $O_2$ and 5% $CO_2$. All cell lines are regularly tested negative for mycoplasma contamination.

The MRC5-SV + Sh cells were obtained by transduction of lentiviral particles produced (as described https://www.addgene.org/protocols/plko/#Ehttps://www.addgene.org/protocols/plko/ - E) from piSMART hEF1α/turboGFP (Dharmacon) doxycycline-inducible lentiviral system containing a Short Hairpin (Sh) Scramble (VSC6572), Sh5-SMN (V3IHSHEG_4923340; mature antisense: TAAACTACAACACCC TTCT) or Sh6-SMN (V3IHSHEG_5297527; mature antisense: TTCAA ATTTTCTCAACTGC). ShSMN both target telomeric *SMN1* and centromeric *SMN2* copies of the gene. The cells were cultured in DMEM supplemented with 10% FBS and 1% P/S; maintained in 100 ng/ml puromycin. Sh expression is induced with 100 ng/ml doxycycline, at least 3 days before experiments. The cells were incubated at 37 °C with 20% $O_2$ and 5% $CO_2$.

### Treatment

DNA damage was inflicted by UV-C light (254 nm, 6-Watt lamp). Cells were globally irradiated with a UV-C dose of 16 J/m$^2$ or locally irradiated with a UV-C dose of 100 J/m$^2$ through a filter with holes of 5 µm. After irradiation, cells were incubated at 37 °C with 5% $CO_2$ for different periods of time.

The PRMT inhibitor, MS023 (ab223596) diluted in DMSO, was added at 1 µM in the medium 20 h before irradiation. The PRMT1 inhibitor, Furamidine (ab287098) diluted in DMSO, was added at 1 µM in the medium 15 h before irradiation.

### Clonogenic assay

Cells were plated at 100 cells per 10-cm dishes. For UV survival experiments, cells were exposed one day after plating to different UV-C doses. Each time point was done in triplicate. 10 days after treatment, the number of clone in each 10-cm dish was counted.

### Transfection of small interfering RNAs (siRNAs)

Cells were seeded in six-well plates with coverslip and allowed to attach. Cells were transfected two times in an interval of 24 h with siRNA using Lipofectamine® RNAiMAX reagent (Invitrogen; 13778150) according to the manufacturer' protocol. Experiments were performed between 24 h and 72 h after the second transfection. Protein knock-down was confirmed for each experiment by western blot. The small interfering RNA (siRNAs) used in this study are: siMock, Horizon, D-001206–14 (10 nM); siCOILIN, Horizon, M-019894-01 (5 nM); siFBL, Horizon, L-011269-00 (10 nM); siPRMT1, Horizon, L-010102-00 (10 nM). The final concentration used for each siRNA is indicated in parentheses. All siRNA are a pool of four different siRNA.

### Protein extraction

Cells were collected using trypsin or by scraping and centrifuged 10 min at 800 g. To verify siRNA efficiency, the coverslip need for the experiment was displaced before fixation and cells that remained in the dish were collected. The extraction of total proteins was performed using the PIERCE RIPA buffer (Thermo, #89900) complemented with EDTA-free cOmplete PIC (ROCHE).

### Western blot

Proteins were separated on a SDS-PAGE gel composed of bisacrylamide (37:5:1), and then transferred onto a polyvinylidene difluoride membrane (PVDF, 0.45 µm; Millipore). The membrane was blocked in PBS-T (PBS and 0.1% Tween 20) with 5% milk and incubated for 2 h at room temperature (RT) or overnight at 4 °C with the primary antibodies diluted in milk PBS-T (see table of antibody).

Subsequently, the membrane was washed with PBS-T (3 × 5–10 min) and incubated with the following secondary antibody diluted 1/5000 in milk PBS-T: Goat anti-rabbit IgG HRP conjugate (170-6515; BioRad) or Goat anti-mouse IgG HRP conjugate (170-6516; BioRad). After the same washing procedure, protein bands were visualized via chemiluminescence (ECL Enhanced Chemo Luminescence; Pierce ECL Western Blotting Substrate) using the ChemiDoc MP system (BioRad).

### Cytostripping

To improve the nuclear signal of immunofluorescence signal, the cytoplasm of the cells was removed before fixation. After two washes with cold PBS, cells were incubated on ice 5 min with cold cytoskeleton buffer (10 mM PIPES pH6,8; 100 mM NaCl; 300 mM Sucrose; 3 mM MgCl2; 1 mM EGTA; 0.5% Triton-X100) followed by 5 min with cold cytostripping buffer (10 mM Tris HCL pH7,4; 10 mM NaCl; 3 mM MgCl2; 1% Tween 40; 0.5% sodium deoxycholate). After three gentle washes with cold PBS, cells were fixed.

### Immunofluorescence

Cells were grown on coverslips, washed with PBS at RT, and fixed with 2% paraformaldehyde (PFA) for 15 min at 37 °C. Cells were permeabilized with PBS 0.1% Triton X-100 (3X short + 2 × 10 min washes). Blocking of non-specific signals was performed with PBS+ (PBS, 0.5% BSA, 0.15% glycine) for at least 30 min. Then, coverslips were incubated with primary antibody diluted in PBS+ for 2 h at room temperature (RT) or overnight at 4 °C in a moist chamber. After several washes with PBS + 0.1% Triton X-100 (3X short + 2 × 10 min) and a quick washed with PBS+, cells were incubated for 1 h at RT in a moist chamber with the following secondary antibody coupled to fluorochrome and diluted 1/400 in PBS+: Goat anti-mouse Alexa Fluor® 488 [A11001, Invitrogen] or 594 [A11005] and Goat anti-rabbit Alexa Fluor® 488 [A11008] or 594 [A11012]. After the same washing procedure but with PBS, coverslips were finally mounted using Vectashield with DAPI (Vector Laboratories).

### Proximity ligation assay

PLA experiments were done using Duolink™ II secondary antibodies and detection kits (Sigma-Aldrich, #DUO92002, #DUO92004, and #DUO92008) according to the manufacturer's instructions. In brief, cells were fixed and permeabilized with the same procedure as immunofluorescence followed by incubation in PLA blocking buffer for 1 h at 37 °C. After blocking, cells were incubated overnight at 4 °C with primary antibodies diluted in PLA Antibody Diluent. After washes with PLA buffer A (1 short + 3 × 5 min), cells were incubated with PLUS and MINUS PLA probes for 1 h at 37 °C. After the same washing procedures with PLA buffer A, if probes were in close proximity (<40 nm), they were ligated together to make a closed circle thanks to the incubation of 30 min at 37 °C with the Duolink™ ligation solution. Then, after the same washing procedures, the DNA is amplified and detected by fluorescence 594 thanks to the incubation of 100 min at 37 °C with the Duolink™ amplification solution. After washing with PLA buffer B (1 short + 2 × 10 min), coverslips were mounted using Vectashield with DAPI (Vector Laboratories).

### RNA fluorescence in situ hybridization (RNA-FISH)

Cells were grown on coverslips and globally irradiated for different times. Then, cells were washed with PBS at RT, and fixed with 4% PFA for 15 min at 37 °C. After two washes with PBS, cells were permeabilized with PBS + 0.4% Triton X-100 for 7 min at 4 °C. Cells were washed rapidly with PBS before incubation for at least 30 min with pre-hybridization buffer (15% formamide in 2X SSPE pH8.0 [0.3 M NaCl, 15.7 mM NaH$_2$PO$_4$.H$_2$O and 2.5 mM EDTA]). 35 ng of probe was diluted

in 70 μl of hybridization mix (2X SSPE, 15% formamide, 10% dextran sulfate and 0.5 mg/ml tRNA). Hybridization of the probe was conducted overnight at 37 °C in a humidified environment. Subsequently, cells were washed twice for 20 min with prehybridization buffer, then once for 20 min with 1X SSPE. After extensive washing with PBS, the coverslips were mounted with Vectashield containing DAPI (Vector Laboratories). The probe sequence (5' to 3') is Cy5- AGACGA-GAACGCCTGACACGCACGGCAC. At least 30 cells were imaged for each condition of each cell lines.

## Primary antibodies

Primary antibodies used are available in Supplementary Information.

## Images acquisition and analysis

Confocal images of the cells were obtained on a Zeiss LSM 880 NLO confocal laser scanning inverted microscope using either a Plan-Apochromat 63x/1.4 or 40x/1.3 oil immersion objective. Other images were obtained using an upright Zeiss Z1 Imager with a Plan-Apochromat 40x/0.95 objective. Time-lapse imaging of live cells for 48 h was achieved on a Zeiss AxioObserver 7 microscope with an Alpha Plan-Apochromat 63x/1.46 oil objective, equipped with a dedicated incubator system set at 37 °C and 5% $CO_2$.

Images of the cells for each experiment were obtained with the same microscopy system and constant acquisition parameters. Images were analyzed with Image J software. For all images of this study, nuclei and nucleoli were delimited with dashed and dotted lines respectively, using DAPI staining.

Protein localization data represented as composite bar graphs was performed by manually sorting cells based on protein localization. The number of cells analyzed per time point or condition was between 100 and 300 cells, except for primary cells, motoneuons and GFP-SMN expressing cells (50 to 100 cells).

## Statistics and reproducibility

All experiments have been performed independently at least three times (biological replicates) with similar results. Error bars represent the Standard Error of the Mean (SEM) of one replicate. Excel was used for statistical analysis and plotting of all the numerical data. Statistics were performed using a Student's test to compare two different conditions (siMock vs. siRNA X or No UV irradiation vs. after irradiation) with the following parameters: two-tailed distribution and two-sample unequal variance (heteroscedastic).

## Reporting summary

Further information on research design is available in the Nature Portfolio Reporting Summary linked to this article.

# Data availability

Source data generated in this study are provided in the Source data file. Source data are provided with this paper.

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

## Acknowledgements

Dedicated to the mothers and fathers of children affected with SMA. We would like to thank Laura Trinkle-Mulcahy for the generous gift of the mCherry-N1. For access to the confocal microscope, we acknowledge the contribution of the imaging facility CIQLE [https://ciqle.univ-lyon1.fr] (Centre d'imagerie Quantitative Lyon-Est - SFR Santé Lyon-Est, UAR3453 CNRS, US7 INSERM, UCBL), a member of LyMIC (Lyon Multiscale Imaging Center). This work was supported by AFM-Téléthon (MyoNeurALP 1 and 2) and by Institut National du Cancer (InCa, PLBIO19-126). S.M. was funded by Jazan university, Jazan, Saudi Arabia. L.M.D. is funded by AFM-Téléthon (MyoNeurALP 2). Z.Z. and J.H. are funded by China Scholarship Council (CSC). P.L. and O.B. are funded by an AFM-Téléthon grant (MyoNeurALP) and by the Joint Collaborative Research Program between the Centre for Neuromuscular Disease (University of Ottawa) and Institut NeuroMyoGène (INMG - Claude Bernard Université Lyon 1). J.C. is funded by a Canadian Institutes of Health Research (CIHR) grant (MOP 123381) and CureSMA Canada. D.M. is a FNRS Research Associate and is funded by grants from the FNRS – Belgium, Fonds Léon Frédéricq (FLF) and Fonds Spéciaux de la Recherche de l'Université de Liège (ULiege).

## Author contributions

S.M., Z.Z., C.M., P.R., O.B., J.H., A.J., L.C. performed experiments. Z.Z., C.M., J.H. quantified the data. L.M.D., G.G.M. performed experiments, analyzed and quantified the data, wrote the article. POM imaged and analyzed the data. P.L., C.M., L.S., D.M., J.C. provided material.

## Competing interests

The authors declare no competing interests.
