## [Peer Review File · Nature Communications]

Nucleolar reorganization after cellular stress is orchestrated by SMN shuttling between nuclear compartmentsREVIEWER COMMENTS

Reviewer #1 (Remarks to the Author):

This manuscript by Musawi and colleagues describes the interaction and subnuclear trafficking of SMN, fibrillarin and coilin in response to DNA damage and repair. This is a very interesting study in that, while other investigators have examined the nucleolar re-localization (displacement) of SMN, fibrillarin and coilin upon DNA damage, the work presented here details how this re-localization of SMN is impacted by fibrillarin and coilin. The authors also examine the repositioning of these three proteins, and the role that fibrillarin and coilin play in SMN localization, when DNA repair is completed.

On balance, this is an excellent manuscript. However, I have a few minor concerns:

1) For the most part, the paper is well-written. However, there are a few awkwardly written sentences and many spelling errors. Before publication, the manuscript should be carefully checked for grammatical errors. For example, tubulin is misspelled in FigS2 D, an accent is present in the word "schematic" in Fig S3 B, repositioned is misspelled in lines 255, 285, and 371 and tumbling is misspelled in line 475. The figure 5 legend heading is confusing. Shouldn't the heading read "The release of SMN from the nucleolus is FBL-dependent"?

These are just a few examples. Again, the manuscript should carefully vetted for grammatical errors.

2) I may have missed it, but how were the cells quantified? In the legend for figure 1 and other legends, + = 50-70% of the representative cells showing a phenotype, ++ = 70-90% and +++ = > 90%. How many cells were counted for each condition? How many biological replicates? This information should be added to the legends and methods.

3) I think the reader would greatly appreciate a model detailing how DNA damage recruits SMN to the nucleolus, and then how SMN is repositioned to CBs after DNA repair. The role of fibrillarin and coilin in these processes should be shown in the model, as should the role of PRMT1.

4) Given the important role of PRMT5 in the modification of fibrillarin and coilin, it struck me as odd that PRMT5 was not mentioned in the Discussion. Would the authors see the same response if PRMT5 was reduced or inhibited compared to PRMT1? If this experiment is not included in the manuscript (and I don't think it needs to be included), then the potential role of PRMT5 in the described processes should be included in the Discussion.

Reviewer #2 (Remarks to the Author):

Musawi et al examine the organization of the nucleolus after stress responses, which has been previously reported and studied. Nucleolar DNA and proteins RNAP1 and fibrillarin are exported to the periphery of the nucleolus until full DNA repair has taken place. Fibrillarin has been involved in restoring a proper nucleolar structure after cellular stresses. The authors focus here on a potential role of SMN protein mainly because it is a fibrillarin interacting protein. SMN complex (SMN, gemin2 to 8 and Unrip) is localized in coilin-positive nuclear bodies that crosstalk and share a number of proteins with nucleoli. It is assumed that SMN sub-nuclear localization participates to SMA pathogenesis. There is no compelling evidence that alterations of sub-nuclear localization of SMN are the cause but rather a simple consequence of a pathological process (SMN deficiency) causing SMA. They described here that: (i) SMN shuttles from CBs to nucleoli during DNA repair; (ii) SMN shuttling is dependent on SMN partners, coilin and fibrillarin; (iii) the proper localization of RNA polymerase I and fibrillarin within the nucleolus after DNA repair is SMN-dependent. The potential role of PRMT1 in SMN shuttling is a novel finding in this study. The manuscript is well written and the confocal images of immunofluorescence are convincing, but the link with any disease phenotype and PRMT1 mechanisms are not shown.

Major concerns

Inhibition of RNA pol I and/or II is accompanied by changes in the organization and composition of

nucleoli, including a nucleolar association of coilin and formation of coilin nucleolar cap-like domains. But not all DNA damage/ genotoxic stresses result into altered coilin distribution. Also, key players of pol I activity relate to translation of RPs. Given the role of nucleoli in ribosome biogenesis and ribosomes in translation, did the authors have excluded that the RNA pol inhibitor used herein does not have an impact on the biogenesis of snRNPs and SMN complex composition? It is of note that inhibitors of translation indeed unexpectedly act on the snRNP biogenesis. There are feedback loops.

What about the localization of snRNAs, snRNP proteins and the other components of SMN complex besides SMN and Gemin5? Why to include Gemin5 while is not a direct interactor of SMN within SMN complex? Gemin5 interacts with SMN through Gemin2 and Gemin3/4. Moreover, Gemin5 is involved in translation. Is translation perturbed when RNA pol I and fibrillarin are mis-localized after DNA repair?

SMA is caused by deletion or mutations of SMN1 gene. The authors showed that in vitro interaction between fibrillarin and SMN is altered by mutations in SMN Tudor domain. What about the effects on nucleolar organization of mutations in the YG domain that have an impact on SMN oligomerization which alter SMN function without alteration of fibrillarin interaction?

The disease is characterized by the loss of spinal motor neurons and muscle atrophy. The results herein are interesting for dividing cells (MRC5 cell line, human embryonic fibroblast-like cells or patient fibroblasts) but they appear preliminary to the field of SMA. Only one SMA patient fibroblast cell line was used. The use of more relevant cell types and/or animal SMA models will strengthen the present work.

As mentioned, some experiments have been performed only twice. It is recommended to perform them at least 3 to 4 times with statistical analyses.

Upon UV-irradiation, they performed GST pull-down using purified recombinant SMN and cellular extracts from UV-irradiated cells. Then, proximity ligation assay (PLA) are used to confirm the results of GST pull-down assay. Some classical co-immunoprecipitation assay could better support the conclusions rather than PLA. In addition, some statistical analyses of the PLA experiments should be included.

Some important results are missing. For siRNA experiments, the authors use « pool » siRNA that are not recommended in cell biology. siRNA molecules should be used individually to reduce the risk of off-targets. Furthermore, rescue experiments should be included to validate the impact of protein depletion.

Labelling of panels in figures. A cell with the genotype *Smn*^{-/-} is not viable. Reduced SMN protein levels should be available. This should be corrected.

The discussion is largely a synthesis of results section, one will have expected some openings about the impact on a better understanding of SMN functions altered in disease. Is there a link between SMN known functions in snRNP biogenesis and inhibition of RNA polymerases? Also, what are the possible mechanisms that could be investigated in future studies?

Reviewer #3 (Remarks to the Author):

Musawi et al., describe a role of SMN in restoring nucleolar organization in response to UV irradiation. They primarily use imaging to look at localization of RNAPI, SMN, FBL, Coilin and Gemin5 within the nucleolus and nucleolar periphery. Most of their phenotypes are changes during the recovery phase (repositioning away from the periphery as induced by UV). By using mainly depletion experiments (siRNA or shRNA) and tracking the displacement and repositioning of RNAPI, FBL and other factors at defined timepoints post UV irradiation. They also show that nucleolar SMN colocalizes with FBL, and that this co-localization is dependent on the presence of Coilin. The interaction between SMN and FBL is further confirmed by co-IP and is reduced for SMN variants containing mutations in the Tudor domain and known to cause SMA. As RNAPI and SMN briefly colocalize both after UV (40 hrs post UV)

and in response to RNAPI inhibition by Cordycepin (2 hrs inhibition and 1 hr recovery after 3 hrs Cordycepin inhibition) they suggest that SMN shuttling into the nucleolus is a general phenomenon after RNAPI inhibition.

The authors make a new and clear connection between SMN shuttling and its functional importance for nucleolar repositioning in response to UV irradiation. On the mechanistic side they show that Coilin and FBL are involved, and RNAPI repositioning is impaired in cells depleted in SMN, Coilin or FBL. One question left largely unaddressed however, is the functional consequence of the delayed RNAPI repositioning. Data presented here suggest that it is not relevant for restart of rRNA synthesis, but it is unclear whether this causes other transcription issues, impacts rDNA repair or translation in response to UV. As mutations within SMN are causative for the autosomal recessive neuromuscular disease SMA, it would be interesting to address SMN mutations causative for SMA results in delayed nucleolar repositioning, as this could be relevant to understand the molecular basis of SMA. Currently it is only shown that SMN mutations impacts the interaction with FBL. It should also be noted that another paper has recently reported that SMN is required both for rRNA synthesis and rDNA repair (Karyka et al., 2022 Life Sci Alliance – referred to as reference 34 in the discussion). As a general comment, the authors could improve the manuscript's readability if they would include more background and clearly state their hypotheses. It is currently unclear why they decided to look at SMN in the context of nucleolar organization, and what in particular that made them look at FBL, coilin and Gemin5. Why these, and not other factors involved in nucleolar organization? The placement of the different figures within the main text and lack numbering within the figures itself (often with the figure legend on a separate page) made it difficult to connect figures and figure legends. I have tried my best and hope I refer to the correct numbering below in my comments.

Major concerns:

1. The conclusions in the manuscript would be greatly supported by time-lapse microscopy of tagged RNAPI for certain central experiments. It is surprising that this is not included as RNAI-GFP stably expressing cell lines has previously been used in the lab of the corresponding author – reference 7.
2. The authors need to report the changes in localization in a more quantitative manner. Currently they use 'The number of representative cells are indicated as followed + : 50–70%; ++ : 70–90%; +++ : >90%'. So when a picture is shown with a + this could an effect just be seen in half of the cells? This is not very informative. A scatterplot/barplot with cells showing co-localization at individual time-points would provide more information.
3. The authors show that SMN protein variants containing mutations within the Tudor domain bind less well to FBL using GST-SMN pull downs and western blotting. How do any of these SMN mutations impact nucleolar organization in response to UV irradiation?
4. The authors have two shRNA targeting SMN (sh5 and sh6). However, for some of the experiments where individual data is shown the two shRNA gives very different results. One looks more like WT and one more like the CSB knockout situation. This makes it impossible to interpret the data, for instance in Fig. 2S.
5. Explain the difference in nucleolar repositioning of transformed fibroblast and primary fibroblasts lacking functional CSB (Fig. 1A vs 1B). Data in Fig. 1C and 1D indicate that both cell lines fail to restart rRNA transcription after UV exposure, but this would suggest that nucleolar repositioning is not relevant for transcription restart. What is it relevant for then? This should be made clearer in the manuscript – and emphasized in the discussion, especially since the lack of effect of rRNA synthesis in cell lacking SMN reported here conflicts with the recent publication Karyka et al., 2022 Life Sci Alliance.
6. The authors say SMN shuttling is important for nucleolar homeostasis. What are the cellular consequences if nucleolar homeostasis is disrupted?
7. It is unclear how SMA causing mutations in SMN relates to the nucleolar reorganization phenotype reported by the authors. If the authors want to make the connection to SMA (which they do at several points throughout the manuscript and in the abstract) they should establish whether nucleolar repositioning is impaired in SMA1^{-/-} cells re-expressing SMA causing SMN mutations.

Minor concerns:

1. How general is this effect to genotoxic agents (a term used by the authors several times in the abstract)? Is SMN mediated nucleolar reorganization important after treatments that would induce genotoxic stress other than UV, such as drugs inducing double-stranded DNA breaks?
2. In Fig. 2, two images are shown for the 40h PUVI for all conditions – but the results for the two images are quite different. One has a '+' and the other not. Does this mean both phenotypes are

observed? It should be made clear both in the main text and the figure legend how representative both situations are.

3. In the methods section (image acquisition and analysis) the authors write: '11 experiments have been performed at least two times and are biological replicates.' What does the 11 refer to? Which experiments is this referring to? All imaging experiments??

POINT-BY-POINT REBUTTAL

In the following rebuttal letter, we will explain how we answered point-by-point to the revisions requested in your decision letter. For simplicity and to increase the readability of the changes we made, the modified text inside the article is highlighted in blue.

REVIEWER 1

POINT 1:

The reviewer asked to revise the grammatical errors.

We thank the reviewer for spotting the different errors and we corrected the grammatical errors indicated and revised the entire article. We also changed the heading of Figure 5 as suggested by the reviewer.

Nevertheless, we would like to indicate that the word "repristinated" is not a misspelling of the word "repositioned" but it is a verb meaning "to restore to the first or original state or condition". We understand this might be a verb that is not often used and for this reason, we changed it in the text with the verb "restore".

POINT 2

The reviewer asks precisions about the quantification of the different IF. "I may have missed it, but how were the cells quantified?"

Indeed, we agreed with the reviewer that the quantifications were not properly discussed and presented: in the first version we used "+" or "-" to indicate which kind of localization was the most or the least present. However, because we thought it would be more informative to give real numbers, we decided to quantify exactly the IF images and present our results of quantification in the principal figures (or in the supplementary figures) as composite bar graphs. In this new version, all IFs and PLAs are accompanied with their corresponding quantifications. These quantifications are the results of several experiments that were repeated and imaged again. The number of cells per quantification ranged generally between 100 to 300 cells per condition, unless specified for certain type of cells (primary cells and motoneurons) or treatments, in which the number of cells was lower. Quantification was performed by sorting and different experimenters have quantified the same images to avoid individual bias. The method of quantification is now described in detail in the "M&M" section (ll. 679-681). Although this method was very time consuming, we think it really adds solidity and clarity to our results

POINT 3:

The reviewer asks for a model that would represent our findings.

Definitely we agree with the reviewer and we add a model as Figure 9 in this new revised version.

POINT 4:

The reviewer ask that we discuss the role of PRMT₅, His questions were:

Would the authors see the same response if PRMT₅ was reduced or inhibited compared to PRMT₁? If this experiment is not included in the manuscript (and I don't think it needs to be included), then the potential role of PRMT₅ in the described processes should be included in the Discussion.

We would like to thank the reviewer for this remark, indeed in the first version the experiment with depletion of PRMT₅ was not included, but we had performed it indeed. We actually, screened for depletion of all PRMTs and just found that the reduction of PRMT₁ and inhibition of the activity of PRMT₁ had an effect on SMN shuttling from CBs to the nucleolus. We agree with the reviewer that this is a very important point and

indeed it was a surprise for us to find that reduction of PRMT5 does not influence the shuttling of SMN. We then decided to include this experiment in this new revised version in Supplementary Figure 13 and the “results” section (ll. 391-396) and to discuss the results in the “discussion” section (ll. 533-536). Briefly, our interpretation of this result is that SMN function in nucleolar homeostasis is independent of the snRNPs biogenesis process.

REVIEWER 2

POINT 1

Reviewer comment: Inhibition of RNA pol I and/or II is accompanied by changes in the organization and composition of nucleoli, including a nucleolar association of coilin and formation of coilin nucleolar cap-like domains. But not all DNA damage/ genotoxic stresses result into altered coilin distribution. Also, key players of pol I activity relate to translation of RPs. Given the role of nucleoli in ribosome biogenesis and ribosomes in translation, did the authors have excluded that the RNA pol inhibitor used herein does not have an impact on the biogenesis of snRNPs and SMN complex composition? It is of note that inhibitors of translation indeed unexpectedly act on the snRNP biogenesis. There are feedback loops.

Indeed, this is a very good point that could be the scope of another full article and would require not only several years of research but also, for us, to acquire expertises that we do not have at the moment to answer these questions. We will look for collaborations in the field of the biogenesis snRNPs in the future to fulfil this arduous task. Momentarily, we believe this point, although important, is beyond the scope of this article.

Nevertheless, we would like to briefly answer that we cannot exclude that UV-irradiation is affecting the biogenesis of snRNPs, nevertheless because UV-irradiation is only blocking RNAP₁ for some hours, we can also not exclude that this effect would be just short in time and probably elusive to detect.

POINT 2

Reviewer comment: What about the localization of snRNAs, snRNP proteins and the other components of SMN complex besides SMN and Gemin5? Why to include Gemin5 while is not a direct interactor of SMN within SMN complex? Gemin5 interacts with SMN through Gemin2 and Gemin3/4. Moreover, Gemin5 is involved in translation. Is translation perturbed when RNA pol I and fibrillarin are mis-localized after DNA repair?

The reviewer here raises a very important point and we must admit, we were in the process of investigate this point thoroughly in another article, but finally we decided to include it in this work, because we agree with the reviewer that this is a very significant aspect to study and discuss. We have studied the localisation of Gemin 2, 3 and 4 during nucleolar reorganisation and found that Gemin 3 and 4 perfectly colocalised with SMN during this process. Gemin 2 was present in both CBs/Gems and Nucleoli also in undamaged cells and its localisation and quantity within the nucleoli increased during the process. Intriguingly, while SMN returns to the CBs/Gems after the repositioning of RNAP₁ within the nucleolus, Gemin 2 remains in the nucleolus longer. Because this result is very important, we presented it in a new principal figure (Figure 3), together with Gemin 5 results. This unexpected result is important because it points to the fact that SMN complex might be slightly different and that the full restoration of the composition of CBs/Gems might be delayed during the DNA damage nucleolar homeostasis process. We presented these new results in the “results” section in the new paragraph “SMN complex shuttles at the nucleolus after UV-irradiation” (ll. 215-234). We discussed this result in the “discussion” section (ll. 498-515).

POINT 3

Reviewer comment: SMA is caused by deletion or mutations of SMN1 gene. The authors showed that in vitro interaction between fibrillarin and SMN is altered by mutations in SMN Tudor domain. What about the effects on nucleolar organization of mutations in the YG domain that have an impact on SMN oligomerization which alter SMN function without alteration of fibrillarin interaction?

We thank the reviewer for this interesting question. To answer this question and combine it with the one proposed by reviewer 3, we decided to complement the inducible shSMN cell lines with either wild-type GFP-SMN and GFP SMN mutants in the Tudor domain and in the oligomerization domain (YG). Because the sh-sequences are raised against the 3' UTR of SMN, the cDNAs from these plasmids are resistant to the sh-reduction. We transfected the plasmids into shSMN06 cell line and performed a clonal selection of GFP-SMN^{WT}, GFP-SMN^{E134K}, GFP-SMN^{Y272C}. We performed IF after UV-irradiation and measured both SMN shuttling and RNAP1 return within the nucleolus.

Briefly, we show that both mutants E134K (Tudor domain) and Y272C (oligomerization domain) present a perturbed shuttling of SMN and importantly in both mutants RNAP1 localization within the nucleolus is not restored at later time points.

Because of the importance of these results, we decided to present it in a new paragraph in the "results" section **"SMN mutants are deficient in nucleolar reorganization after DNA damage induction and repair"** (ll. 342-369). Because we have already a high number of principal figures, we decided to include these results as a supplementary figure that illustrate the results (Figure S11) but if the editors think this result deserves a principal figure, we can of course restructure the layout of the figures.

POINT 4

Reviewer comment: The disease is characterized by the loss of spinal motor neurons and muscle atrophy. The results herein are interesting for dividing cells (MRC5 cell line, human embryonic fibroblast-like cells or patient fibroblasts) but they appear preliminary to the field of SMA. Only one SMA patient fibroblast cell line was used. The use of more relevant cell types and/or animal SMA models will strengthen the present work.

We would like to thank the reviewer for this interesting question. To answer this request, we acquired Neuronal Precursor (derived from WT or SMA patient's iPSC) from Dr Cecile Martinat (iSTEM, Evry) and we differentiated these NP into motoneurons. We then performed the IF of RNAP1 after UV-irradiation. Quantification of these experiments show that RNAP1 does not return within the nucleolus after DNA repair in SMA motoneurons, just as transformed or primary fibroblasts. Results are presented in **Supplementary Figure 2**.

The difficulty of obtaining these NP, the few numbers of fully differentiated motoneurons at the end of the experiment together with the higher sensitivity of these cells to UV did not allow us to perform more experiments, but we plan to find new strategies in the future to be able to work more with SMA derived cells or shSMN inducible iPSC produced in house.

We described the cell culture method of iPSC derived motoneurons in the **supplementary information**.

POINT 5

Reviewer comment: As mentioned, some experiments have been performed only twice. It is recommended to perform them at least 3 to 4 times with statistical analyses.

All experiments were repeated at least 3 times. Quantifications and statistical analysis are presented for each experiment, Statistical method has been described in the **"Mat and Med" section (ll. 682-686)**.

POINT 6

Reviewer comment: Upon UV-irradiation, they performed GST pull-down using purified recombinant SMN and cellular extracts from UV-irradiated cells. Then, proximity ligation assay (PLA) are used to confirm the results of GST pull-down assay. Some classical co-immunoprecipitation assay could better support the conclusions rather than PLA. In addition, some statistical analyses of the PLA experiments should be included.

Unfortunately, co-immunoprecipitations are not compatible with siRNA treatments. In fact, siRNAs are usually used on very small number of cells (100.000 cells) and co-IPs require at least 10 million cells with the antibodies that we have tested. We have tried in the past to scale up the siRNAs transfections to adapt them to biochemical studies, but we failed to have a proper extinction of the protein signal in scaled-up conditions. That is the reason why siRNAs-approaches are almost always used in combination with single cell assays. We than choose PLA as a substitute technique which has also the advantage to show where in the cell the interaction/proximity is taking place. In our case, in fact, most of the interactions take place in the nucleolus and this information is very important for our study. Obviously, with IPs, cellular localization of the interaction cannot be discriminate. We performed quantifications of all the PLAs presented in the article, statistical tests have been performed and are now included in the figures and legends of figures.

POINT 7

Reviewer comment: Some important results are missing. For siRNA experiments, the authors use « pool » siRNA that are not recommended in cell biology. siRNA molecules should be used individually to reduce the risk of off-targets. Furthermore, rescue experiments should be included to validate the impact of protein depletion.

We do not agree with the reviewer as pools of siRNAs are on the contrary designed to reduce off-target effects and the reason why is that in smart pools each siRNAs are less concentrated and can be used at concentrations that are usually not the one causing off-targets effects. In our protocol, we improved the protein extinction by transfecting twice at 24 h interval, this technique allows us to decrease even more the amount of siRNA used, in our assays we reduced this concentration to only 10nM of the smart pool, which corresponds to only 2.5nM of the individual siRNAs. When using individual siRNAs, their final concentration is 10nM for each.

Nevertheless, we think that, as the reviewer, some readers might not be used to the use of siRNAs smart pools, so we decided to show that individual siRNAs show the same biological effects as pools. We ordered the individual siRNAs included in the pools. As expected amongst the pool some siRNAs did not produce any extinction of the proteins and were not used further. We choose for Coilin and Fibrillarin, two individual siRNAs that reduced the protein levels enough (more than 50% reduction) and repeated the experiments and, with no surprise, the individual siRNAs produced the same results as the siRNAs pools. Results of this experiment are presented in Supplementary Figure 9 and 10. We think that showing that 2 individual siRNAs and the pool are sufficient proofs of the validity of our findings. For the experiment involving siPRMT1, we did not think it was necessary to use individual siRNAs, because we used specific drugs to inhibit PRMT1 function.

The sequences of the individual siRNAs used is now given in the supplementary information.

POINT 8

Reviewer comment: Labelling of panels in figures. A cell with the genotype *Smn*^{-/-} is not viable. Reduced SMN protein levels should be available. This should be corrected.

We changed *SMN*^{-/-} into *SMN*^{KD}.

POINT 8

Reviewer comment: The discussion is largely a synthesis of results section; one will have expected some openings about the impact on a better understanding of SMN functions altered in disease. Is there a link between SMN known functions in snRNP biogenesis and inhibition of RNA polymerases? Also, what are the possible mechanisms that could be investigated in future studies?

Discussion was largely improved, partly by the unexpected results obtained from the new experiments we performed and partly because we do show a model and discuss the future studies. **New paragraphs can now be found on ll. 486-521, ll. 522-529 and ll. 533-539.**

REVIEWER 3

General point

Reviewer asks what are the consequences of the delayed RNAP1 repositioning in SMN cells.

This is a very interesting question and we do not have yet a clear answer, simply because we already know that even without a stress SMN cells are deficient for ribosome biogenesis, snRNP production, splicing, and many other functions. In our work, we show that SMN cells are proficient concerning DNA repair of UV-lesions, and that there is not a defect in the DNA repair reaction *per se*. We think that what makes our work different and original is that we show that SMN plays a role in the recovery of a normal situation after DNA repair completion. Nevertheless, the question of the biological consequences of this perturbed function in SMN cells is really relevant, we then decided to perform clonogenic survival after UV-irradiation and importantly after chronic oxidative damage induction and in both cases, we could show that SMN cells have a deficiency in this assay. These results point to the fact that not just the DNA repair process is important for cell viability but also all the processes that restore cellular functions after DNA repair completion.

Because, we think that these results are important, not only for the SMA/SMN community, but also for the DNA repair field, we decided to include a new paragraph in the "results" section which perfectly end the article in our opinion, the paragraph is "**SMN cells are sensitive to DNA damage**" (ll. 434-454). These results are also illustrated in a **principal figure (Figure 8)** and discussed in the "**discussion**" section (ll. 555-564).

Reviewer asks why we were interested in SMN, coilin, fibrillarin and Gemin 5 in the context of the nucleolar reorganisation.

We thank the reviewer for this question. We stated in the introduction (ll. 55-71) how we stumbled upon SMN. In fact, we have previously found (but not yet published) that fibrillarin (together with other proteins) is important for the RNAP1 repositioning after DNA repair. Part of these results are available on biorxiv (<https://doi.org/10.1101/646471>) but we did not publish them (yet) because we did not have a molecular mechanism to explain our findings. In order to disclose this mechanism and convinced that other proteins would be involved in this process, we proceeded to do a small screen by "best candidate" approach and looked into fibrillarin interacting partners. SMN was one of them. Then, of course, once SMN was found to have an effect on RNAP1 repositioning it was important to study the relation between coilin, fibrillarin, Gemins as these proteins are partners of SMN. Of course, our findings are now leading to many more questions to be solved and SMN is a particularly interesting protein to study. We are not expert in this field (yet) and that is why we were maybe reticent in "including more background and clearly state our hypotheses" but we have tried to be more daring in the "Discussion Section" of this new version.

POINT 1

Reviewer comment: The conclusions in the manuscript would be greatly supported by time-lapse microscopy of tagged RNAPI for certain central experiments. It is surprising that this is not included as RNAI-GFP stably expressing cell lines has previously been used in the lab of the corresponding author – reference 7.

We agree with the reviewer that a time lapse would have been thrilling to perform and indeed, our group has a great experience in this field. Nevertheless, there is a very important technical issue when one wants to perform time lapse imaging on nucleoli. As you might know, nucleolus is a stress sensor and imaging cells is a source of stress. Since we imaged for the first time the RNAP1 displacement and repositioning (back into 2018) we tried to image in real-time the whole process. Up to now, we never succeeded to have the full process (which lasts 48 hours), because the number of imaging required to do a movie would be too high and cells would either die or nucleolus would just not recover its normal structure after 48 hours of imaging. It has been really frustrating not to be able to see this process in living cells. Nevertheless, we have taken the suggestion of the reviewer as a challenge and try once again to find the good number of images that will allow us to show for the first time the shuttling of SMN in living cells. We produced a stably expressing Cherry-SMN cell line, by transfecting Cherry-SMN plasmid into the sh6-SMN inducible cell line. Because the sh6 is raised against the 3'UTR of SMN, we could downregulate the endogenous SMN and follow the Cherry-SMN fluorescent version in living cells by imaging once per hour the whole process. Imaging more often would delay the whole process (inducing an extra stress to the cells). This movie was very informative as we could visualize SMN in CBs/Gems in undamaged cells and we could see SMN disappear from CBs/Gems after damage and appear at the periphery of nucleoli, then reappearing at the end of the process in CBs.

Our results are now illustrated in Figure S7 and presented in the “results” section (ll. 194-202).

POINT 2

Reviewer comment: The authors need to report the changes in localization in a more quantitative manner. Currently they use ‘The number of representative cells are indicated as followed + : 50–70%; ++ : 70–90%; +++ : >90%’. So when a picture is shown with a + this could an effect just be seen in half of the cells? This is not very informative. A scatterplot/barplot with cells showing co-localization at individual time-points would provide more information.

Indeed, as stated in Reviewer 1 point 2, we agreed with the reviewer(s) that the quantifications were not very informative.

In the first version we used “+” or “-” to indicate which kind of localization was the most or the least present. However, because we thought it would be more informative to give real numbers, we decided to quantify exactly the IF images and present our results of quantification in the principal figures (or in the supplementary figures) as composite bar graphs. In this new version, all IFs and PLAs are accompanied with their corresponding quantifications. These quantifications are the results of several experiments that were repeated and imaged again. The number of cells per quantification ranged generally between 100 to 300 cells per condition, unless specified for certain type of cells (primary cells and motoneurons) or treatments, in which the number of cells was lower. Quantification was performed by sorting and different experimenters have quantified the same images to avoid individual bias. The method of quantification is now described in detail supplementary information. Although this method was very time consuming, we think it really adds solidity and clarity to our results

POINT 3 and POINT 7

Reviewer comment 3: The authors show that SMN protein variants containing mutations within the Tudor domain bind less well to FBL using GST-SMN pull downs and western blotting. How do any of these SMN mutations impact nucleolar organization in response to UV irradiation?

Reviewer comment 7: It is unclear how SMA causing mutations in SMN relates to the nucleolar reorganization phenotype reported by the authors. If the authors want to make the connection to SMA (which they do at several points throughout the manuscript and in the abstract) they should establish whether nucleolar repositioning is impaired in SMA1^{-/-} cells re-expressing SMA causing SMN mutations.

We thank the reviewer for this interesting question. To answer this question and combine it with the one proposed by reviewer 2, we decided to complement the inducible shSMN cell lines with either wild-type GFP-SMN and GFP SMN mutants in the Tudor domain and in the oligomerization domain (YG). Because the sh-sequences are raised against the 3' UTR of SMN, the cDNAs from these plasmids are resistant to the sh-reduction. We transfected the plasmids into shSMN₆ cell line and performed a clonal selection of GFP-SMN^{WT}, GFP-SMN^{E134K}, GFP-SMN^{Y272C}. We performed IF after UV-irradiation and measured both SMN shuttling and RNAP1 return within the nucleolus.

Briefly, we show that both mutants E134K (Tudor domain) and Y272C (oligomerization domain) present a perturbed shuttling of SMN and importantly in both mutants RNAP1 localization within the nucleolus is not restored at later time points.

Because of the importance of these results, we decided to present it in a new paragraph in the "results" section **"SMN mutants are deficient in nucleolar reorganization after DNA damage induction and repair" (II. 342-369)**. Because we have already a high number of principal figures, we decided to include these results as a supplementary figure that illustrate the results (**Figure S11**) but if the editors think this result deserves a principal figure, we can of course restructure the layout of the figures. Construction and expression of GFP-SMN and mutants is now described in the **supplementary information**.

POINT 4

Reviewer comment: The authors have two shRNA targeting SMN (sh5 and sh6). However, for some of the experiments where individual data is shown the two shRNA gives very different results. One looks more like WT and one more like the CSB knockout situation. This makes it impossible to interpret the data, for instance in Fig. 2S.

In our study, we consistently verified each of the shSMN and siRNAs for each of our experiments and of course we noticed certain variations in the reduction of signal. We only considered the results when the reduction was of at least 80% and discard the experiments where this level was lower. It has also to be noted that the shSMN 5 and 6 are lentiviral transfected population of cells, we never thought to do a clonal selection, until very recently, when we noticed that in this population some SMN positive cells were still present (estimated at 5%). In brief, different results are due to the shRNA efficiency and the presence of a certain percentage of SMN positive cells.

Nevertheless, for the specific data mentioned by the reviewer, we can state that for what concern the DNA repair tests (now all in supplementary figure S4), the two sh cell lines behave always more like WT than CSB knockout. In this kind of assays what matters is the recovery (or not) of the cellular activity tested before induction of the damage. It has been often recorded that some cells will recover more than the wild-type or more than a similar cell type. However, DNA repair deficient cells will not recover at all the activity of RRS, RNA Fish, TCR-UDS or UDS. In our assays, although sh6 and sh5 might have different values between each other or compared to the wild type cells, they both recover their cellular activities and their values are always statistically different than the DNA repair deficient cell line used in the test.

POINT 5

Reviewer comment: Explain the difference in nucleolar repositioning of transformed fibroblast and primary fibroblasts lacking functional CSB (Fig. 1A vs 1B). Data in Fig. 1C and 1D indicate that both cell lines fail to restart rRNA transcription after UV exposure, but this would suggest that nucleolar repositioning is not relevant for transcription restart. What is it relevant for then? This should be made clearer in the manuscript – and emphasized in the discussion, especially since the lack of effect of rRNA synthesis in cell lacking SMN reported here conflicts with the recent publication Karyka et al., 2022 Life Sci Alliance.

Indeed, RNAP1 transcription can restart from the periphery of the nucleolus, we already showed this in Daniel et al 2018 PNAS. In this article we show that in XPC cells (which repair UV-lesions on transcribed genes but do not repair UV-lesions in the rest of the genome) RNAP1 transcription restarts but because UV-lesions are still present in untranscribed regions on the nucleolar DNA, the “signal” for RNAP1 to remain at the periphery of the nucleolus is still present. This RNAP1 transcription is taking place in a non-canonical structure and the consequence might not be directly related to the ability of producing 47S but more on the processing of the 47S (modifications, maturation, export). In SMN cells, we demonstrate that UV-lesions are repaired both in transcribed (TCR-UDS, RNA-Fish) and untranscribed regions of the genome (UDS). UV-lesions are repaired by the Nucleotide Excision Repair. In the paper Karyka et al., 2022 Life Sci Alliance the rDNA problems are observed because of the accumulation of R-loops and the inability/deficiency in SMN cells to process these DNA:RNA hybrids which are very frequent in rDNA transcribed genes. The damage induced system in our study is different because lesions are different and the involved DNA repair pathway is also different. So to the question what is “nucleolar repositioning relevant for ?” the answer is probably the next steps of ribosome biogenesis and we demonstrate that this defect in SMN cells leads to a certain sensitivity to damage, both UV-lesions and chronic oxidative damage. This will be explained in our next answer (POINT6).

POINT 6

Reviewer comment: the authors say SMN shuttling is important for nucleolar homeostasis. What are the cellular consequences if nucleolar homeostasis is disrupted?

This question is a very important one and we are happy to have been able to show that consequently to the disruption of nucleolar homeostasis we observe a sensitivity to DNA damage, that was not shown before for SMN cells. In order to highlight this sensitivity to DNA damage, we performed clonogenic survival after UV-irradiation and after chronic oxidative damage and in both cases, we could show that SMN cells have a deficiency in this assay. These results point to the fact that not just the DNA repair process is important for cell viability but also all the processes that restore cellular functions after DNA repair completion.

Because, we think that these results are important, not only for the SMA/SMN community, but also for the DNA repair field, we decided to include a new paragraph in the “results” section which perfectly end the article in our opinion, the paragraph is “**SMN cells are sensitive to DNA damage**” (ll. 434-454). These results are also illustrated in a **principal figure (Figure 8)** and discussed in the “**discussion**” section (ll. 555-564).

Minor Point 1

Reviewer comment: How general is this effect to genotoxic agents (a term used by the authors several times in the abstract)? Is SMN mediated nucleolar reorganization important after treatments that would induce genotoxic stress other than UV, such as drugs inducing double-stranded DNA breaks?

In order to induce another type of stress/damage, we decided to concentrate to the oxidative damage, which is very common in cells and is produced endogenously by cellular metabolism. There are two ways of inducing oxidative damage: (i) an acute dose of potassium bromate (KBrO₃) which is quickly repaired by the Base Excision repair system, (ii) a chronic exposure by culturing cells at 20% of O₂, this concentration although very common

in cell culture procedures is considered to be a hyperoxia condition compared to normoxia observed in tissues, which range is 3-5% O₂. We cultured our cells in normoxia as we avoid to induce oxidative damage. To induce chronic oxidative damage, we culture SMN cells in 20% of O₂. Interestingly, we found that SMN cells are sensitive to chronic oxidative damage as stated in POINT 6. We also checked whether an acute dose of oxidative damage would cause the shuttling of SMN and indeed that is the case. We have illustrated these results in supplementary Figure S6A. The treatment with KBrO₃ is now described in supplementary information. So, we think that the shuttling of SMN is a common feature for at least these two types of damage and other cellular stresses, like RNAP1 transcriptional block as observed in Figure S6 B and S6C.

Minor Point 2

Reviewer comment: In Fig. 2, two images are shown for the 40h PUVI for all conditions – but the results for the two images are quite different. One has a '+' and the other not. Does this mean both phenotypes are observed? It should be made clear both in the main text and the figure legend how representative both situations are.

The time 40 hours is here an "In between situation". Because it is not very informative and in order to save space, we have excluded it from the principal figures and kept it in some Supplementary figures. Quantification of all the time points have been conducted and precise percentages of different groups of cells are now presented for each experiment.

Minor Point 3

Reviewer comment: In the methods section (image acquisition and analysis) the authors write: '11 experiments have been performed at least two times and are biological replicates.' What does the 11 refer to? Which experiments is this referring to? All imaging experiments??

All experiments were repeated at least 3 times. Quantifications and statistical analysis are presented for each experiment, Statistical method has been described in the Mat and Med section (ll. 682-686).

REVIEWER COMMENTS

Reviewer #1 (Remarks to the Author):

The authors have addressed my previous concerns. I appreciate the amount of extra work spent to quantify the IF and PLA data. I also appreciate learning a new Scrabble word (repristinated).

Reviewer #2 (Remarks to the Author):

In their revised manuscript, Musawi et al have made efforts to tackle some of the referee's original concerns. They examine the dynamic behaviour of nucleoli and their composition changes under stress conditions. However, the issues below still are significant concern, either because they were not correctly addressed or because they arise from new data or new discussion.

1. SMN complex being a key factor in snRNP biogenesis, the manuscript should include some snRNP markers. Sm proteins entering the nucleus are also found transiently associated with nucleoli (Sleeman & Lamond, Curr Biol 1999). Sm proteins interact directly with SMN protein. Also, U6 snRNA has nucleolar localization during its life cycle.
2. Because following SMN siRNA, the protein levels of most Gemins are significantly reduced compared with their levels in control cells (Feng et al HMG 2005), the levels of Gemins should be examined and rescue experiments performed.
3. Line 120. How do the cells survive if the RNAPI does not return to the nucleolus? Is there an increased cell death?
4. New data for Gemin2 experiments lead the authors to state "After damage induction, the proportion of CBs/Gems containing Gemin2 decreased and concomitantly the nucleolar localization increased"(line 227, Fig3 and SMN in fig 2D). Gemin2 interacts with RAD51 but SMN does not. Gemin2 enhances RAD51-DNA complex formation by inhibiting RAD51 dissociation from DNA, and thereby stimulates DNA repair (Takizawa et al, NAR 2010). This is a major DNA repair pathway. The authors should consider at least this possibility and being less speculative about Gemin2/SMN (Discussion, line 507). Also, the role of SMN complex in snRNP biogenesis is clearly defined for the cytoplasmic assembly of Sm proteins on pre-snRNAs. In CBs, SMN complex activities are not as well defined yet. Gems are believed to be storage nuclear bodies.
5. Line 555. The authors wrote, "We believe that our study will also open new avenues of research that could improve the well-being of SMA patients". It is not clear how this study could have impact on individuals with SMA and/or on advices for medical management. The study is fundamental biology. The paragraph should be rephrased.
6. The referee notes that all uncropped immunoblots should be found in the supplemental file. In addition, protein marker molecular size should be included in uncropped immunoblot data.
7. There are seven additional authors. The authors contribution should be included.

Reviewer #3 (Remarks to the Author):

The authors have addressed all my concerns adequately. With the added experiments and changes to image quantifications, the revised version of the manuscript is in my opinion now suitable for Nature Communications.

Minor comment: The main figure 8 is for some reason also present in the end of the supplementary material. This must be a simple mistake.

POINT-BY-POINT REBUTTAL

In the following rebuttal letter, we will explain how we answered point-by-point to the revisions requested. For simplicity and to increase the readability of the changes we made, the modified text is highlighted in blue.

REVIEWER 2

POINT 1:

1. SMN complex being a key factor in snRNP biogenesis, the manuscript should include some snRNP markers. Sm proteins entering the nucleus are also found transiently associated with nucleoli (Sleeman & Lamond, *Curr Biol* 1999). Sm proteins interact directly with SMN protein. Also, U6 snRNA has nucleolar localization during its life cycle.

We thank the reviewer for this question. We could gather antibodies for all Sm proteins in a very rapid time, thanks to our collaborator (Denis Mottet). We tested them all in our immunofluorescence conditions. We had workable signal and localisation for snRNP B/B', snRNPF and "Y12" (recognising snRNP B/B', snRNP D1 and snRNP D3). The other antibodies (snRND1, snRNPD2, snRNP D3, snRNPE and snRNPG) did not give a good signal in IF, with too much non-specific signal for some or very faint signal for others (different concentrations were tested).

We explored the possibility that Sm proteins would also shuttle at the nucleolus after DNA damage induction, as suggested by the reviewer, and indeed we could detect their shuttling. Without any damage, Sm proteins are visualised in what we believe are splicing speckles, after DNA damage induction. They do accumulate at the periphery of the nucleolus with a similar initial kinetics than SMN, nevertheless, their return to a normal localisation (within splicing speckles) is not yet completed at 48h. Moreover, we could demonstrate that this change in localisation is not SMN dependent, as no particular change was observed in shSMN cells.

We decided to include this experiment in this new revised version in Supplementary Figure S9 and the "results" section (ll. 245-252). Briefly, our interpretation of this result is that Sm proteins can shuttle at the nucleolus after DNA damage, but this shuttling is not SMN dependent.

POINT 2:

2. Because following SMN siRNA, the protein levels of most Gemins are significantly reduced compared with their levels in control cells (Feng et al *HMG* 2005), the levels of Gemins should be examined and rescue experiments performed.

We thank the reviewer for this question. We have analysed by western blot the steady state levels of Gemin2, 3, 4, 5, 6, 7 and 8, in presence and absence of SMN. Of all the antibodies tested, just Gemin 6 was not visualisable in WB. We determined that, as suggested by the reviewer, Gemin2, 3, 4, 7, and 8 cellular concentration is SMN dependent, namely when SMN is depleted, the amount of these Gemins is decreased and when SMN concentration is rescued with a GFP-SMN expressing construct, the concentration of Gemins is increased. Gemin5 concentration is not modified in SMN Knocked down cells.

We decided to include this experiment in this new revised version in Supplementary Figure S8 and the "results" section (ll. 238-244).

POINT 3:

3. Line 120. How do the cells survive if the RNAP1 does not return to the nucleolus? Is there an increased cell death?

As described in Figure 1D, although RNAP1 does not return in the nucleolus, RNAP1 transcription restarts from a site (the periphery of the nucleolus) which is not the canonical site of RNAP1 transcription. This transcription insures a certain viability for the cells. However, as discussed in the paragraph **"SMN cells are sensitive to DNA damage"** and shown in Figure 8A, SMN cells show a sensitivity to DNA damage which means that there is an increased cell death after DNA damage in SMN cells. As we state in the "discussion" section (ll. 578-580) "This result is also important because they point to the fact that not just the DNA repair process is important for cell viability but also all the processes that restore cellular functions after DNA repair completion"

POINT 4:

4. New data for Gemin2 experiments lead the authors to state "After damage induction, the proportion of CBs/Gems containing Gemin2 decreased and concomitantly the nucleolar localization increased"(line 227, Fig3 and SMN in fig 2D). Gemin2 interacts with RAD51 but SMN does not. Gemin2 enhances RAD51–DNA complex formation by inhibiting RAD51 dissociation from DNA, and thereby stimulates DNA repair (Takizawa et al, NAR 2010). This is a major DNA repair pathway. The authors should consider at least this possibility and being less speculative about Gemin2/SMN (Discussion, line 507). Also, the role of SMN complex in snRNP biogenesis is clearly defined for the cytoplasmic assembly of Sm proteins on pre-snRNAs. In CBs, SMN complex activities are not as well defined yet. Gems are believed to be storage nuclear bodies.

RAD51 is one of the first proteins recruited on Double Strand Breaks and it marks the beginning of the homologous recombination pathway of DNA repair. Each DNA repair pathway is specific for a type of damage and UV-lesions, the only damage induced by UV-irradiation, is repaired by Nucleotide Excision Repair, which is a different pathway that does not repair double strand breaks. Conversely, Homologous Recombination does not repair UV-lesions. There is no apparent reason to believe that RAD51 (exclusively working in HR) would be implicated in UV-lesions repair.

The only paragraph about Gemin2 in the discussion is:" While SMN/Gemin2 foci disappeared during RNAP1 displacement, the nucleolar signal of Gemin 2 increased reaching the maximum after RNAP1 repositioning and Gemin2 foci did not reappear. This result might suggest that SMN complex is not exactly the same in CBs/Gems when performing snRNPs assembly and in the nucleolus during DNA repair-induced nucleolar reorganization. These results also suggest that the restoration of fully functional CBs/Gems containing Gemin2 might be delayed in time, probably further delaying the restoration of the other functions of the SMN complex".

Because this paragraph seems to be problematic for the reviewer, we propose to change it with: "Interestingly, while SMN/Gemin2 foci disappeared during RNAP1 displacement, the nucleolar signal of Gemin 2 increased reaching the maximum after RNAP1 repositioning and Gemin2 foci did not reappear. It is still unknown what retains Gemin2 within the nucleolus at late time points and this retention might influence the reconstitution of nuclear SMN complex" now in the "discussion" section (ll. 523-527).

POINT 5:

5. Line 555. The authors wrote, "We believe that our study will also open new avenues of research that could improve the well-being of SMA patients". It is not clear how this study could have impact on individuals with SMA and/or on advices for medical management. The study is fundamental biology. The paragraph should be rephrased.

We deleted this sentence.

POINT 6:

6. The referee notes that all uncropped immunoblots should be found in the supplemental file. In addition, protein marker molecular size should be included in uncropped immunoblot data.

Our research data (uncropped immunoblot, spreadsheets and microscopy images) will be available in figshare repository.

POINT 7:

7. There are seven additional authors. The authors contribution should be included.

We added a section Author Contribution after the acknowledgements (ll. 712-717).

REVIEWERS' COMMENTS

Reviewer #2 (Remarks to the Author):

The authors have addressed my previous concerns. I appreciate the extra work and additional information.

Minor comments :

1)line 97. Reference #18 is not properly cited.

2)line 766. Primary antibodies Table. A simple mistake took place: Anti-gemin6, 7 and 8 are missing.